# GEMTELLIGENCE: Accelerating gemstone classification with deep learning
Tommaso Bendinelli [1] ✉, Luca Biggio[1,2], Daniel Nyfeler[3], Abhigyan Ghosh[3], Peter Tollan[3], Moritz Alexander Kirschmann[1] & Olga Fink [4]

The value of luxury goods, particularly investment-grade gemstones, is influenced by their origin and authenticity, often resulting in differences worth millions of dollars. Traditional methods for determining gemstone origin and detecting treatments involve subjective visual inspections and a range of advanced analytical techniques. However, these approaches can be time-consuming, prone to inconsistencies, and lack automation. Here, we propose GEMTELLIGENCE, a novel deep learning approach enabling streamlined and consistent origin determination of gemstone origin and detection of treatments. GEMTELLIGENCE leverages convolutional and attention-based neural networks that combine the multi-modal heterogeneous data collected from multiple instruments. The algorithm attains predictive performance comparable to expensive laser-ablation inductively-coupled-plasma mass-spectrometry analysis and expert visual examination, while using input data from relatively inexpensive analytical methods. Our methodology represents an advancement in gemstone analysis, greatly enhancing automation and robustness throughout the analytical process pipeline.

Gemstones, both natural and synthetic, are highly prized for their rarity and beauty and are commonly used in jewelry for both adornment and investment purposes. Some of these minerals can be worth over a million dollars per gram, making them some of the most concentrated physical capital in the world. In addition to factors such as species and aesthetic quality, the value of a gemstone is also influenced by its geographic origin and any potential treatments it may have undergone after being mined. Identifying these treatments, which can include exposure to electromagnetic radiation[1], heating[2,3], or the infusion of oils or other substances[4], is crucial for determining the true value of a precious stone and upholding consumer trust in the jewelry industry. Unfortunately, the artisanal nature of gemstone mining results in a fragmented and opaque supply chain, making it difficult to reliably track the origin and treatment of individual stones.

To minimize investment risks, buyers and sellers often require that gemstones are accompanied by an independent laboratory report that confirms the physical characteristics, treatment status, and geographic source of the stones. In the conventional practice of determining the authenticity and source of gemstones, skilled human experts with two to six years of training in gemology play a pivotal role[5]. These specialists have traditionally relied on optical microscopy to identify structures and inclusions within the gemstones that can provide clues about their origin and any treatments they may have undergone. However, this task is exceptionally difficult, as gemstones from different locations may exhibit strikingly similar features due to shared geological histories[6]. Furthermore, with the advancement of treatment techniques for gemstones, the detection of such treatments has become increasingly challenging for even the most experienced human examiners[7,8]. Consequently, relying solely on visual inspection for reliable and reproducible treatment detection and origin determination is generally considered insufficient in modern settings[9-11].

In response to these pressing challenges and industry demands, state-of-the-art gemology laboratories have introduced a range of analytical instruments into their regular workflow[12]. These include ultraviolet-visible-near-infrared spectroscopy (UV)[13], Fourier-transform infrared spectroscopy (FTIR)[14], energy-dispersive X-ray fluorescence (XRF)[15], and laser-ablation inductively-coupled-plasma mass spectrometry (ICP-MS)[16]. The UV, FTIR, and XRF devices, with a combined cost of around 200,000 USD, can be operated by technical staff with just a brief introductory training. However, the ICP-MS instrument alone, while providing the most comprehensive data for identifying origins, costs approximately 500,000 USD and requires one or more qualified operators with extensive theoretical and practical training. Additionally, ICP-MS uses a laser to ablate a small volume from the gems, making it, therefore, destructive on a micro-scale level[17]. Despite all these data sources, accurately determining the origin and detecting treatments remain highly challenging

[1]CSEM SA, Untere Gründlistrasse 1, 6055 Alpnach Dorf, Switzerland. [2]ETH, Universitätstrasse 6, 8006 Zürich, Switzerland. [3]Gübelin Gem Lab, Maihofstrasse 102, 6006 Lucerne, Switzerland. [4]EPFL, Intelligent Maintenance and Operations Systems, Station 18, 1015 Lausanne, Switzerland. ✉e-mail: tommaso.bendinelli@csem.ch

due to subtle differences in physical, spectroscopic, and chemical properties between stones from different origins[12]. Nonetheless, ensuring the accuracy and reliability of gemstone classification is of utmost importance for the entire field. Indeed, as gemstones are long-term investments, they re-enter the market from time to time, and new evaluations of the same stones are often conducted. Inconsistencies in the origin or treatment determination of the same stone over time, which substantially affect a stone's value, can undermine the confidence in the entire asset class. Strict laboratory protocols, such as restricting experts' access to analytical results during microscopic examinations, ensuring at least two independent conclusions, and usage of ICP-MS are typically in place to reduce subjective biases and ultimately minimize the risk of making an incorrect classification. While these approaches can help in reducing the risk of errors, they also result in substantial cost and time. For this reason, the advancement of methods that effectively leverage data obtained from affordable instruments while maximizing accuracy and robustness is of considerable practical significance.

Modern machine- and deep-learning algorithms have revolutionized the analysis and interpretation of large and complex datasets in various fields, including but not limited to material science[18–20], geoscience[21,22], and computational chemistry[23–25] allowing for more accurate and efficient data processing. However, their application to gemology is still in its infancy. Conventional machine-learning techniques in this field, which typically involve feature extraction methods followed by basic classification algorithms, have provided promising results in automating various gemological tasks, such as categorizing gemstones by type and shape[26,27], distinguishing real and synthetic gemstones[28], and even more complicated tasks like grading gemstones[29]. Nevertheless, such techniques are restricted to analyzing only one type of data source at a time, such as images, spectra, or tabular data[30], limiting their capability of detecting artificial treatments or correctly identifying the origin of the gemstone. As such, these challenging tasks still heavily rely on human expertise.

Herein, we propose GEMTELLIGENCE (Fig. 1), a deep learning-based method that automates the determination of the country of origin

(OD) and detection of treatment (TD) of gemstones at a fraction of the time and cost of professional gemological laboratories, without compromising accuracy. This study is the first of its kind to address both OD and TD of valuable gemstones using a novel deep learning approach specifically tailored to handle varied and multi-modal analytical data acquired from different testing devices. Crucially, we conduct our experimental evaluation on a large collection of high-quality gemstones thereby, allowing us to examine the performance of our algorithm in real-world scenarios. Particularly, we focus on blue sapphires, which are among the "big 3" gemstone species most frequently assessed by top-tier gemology laboratories and also the most challenging to identify[11]. A part of the aforementioned data, along with the source code of our model, will be made available for public use to facilitate the benchmarking and reproducibility of the results presented in this work.

The primary innovation of the proposed approach lies in its multi-modal design, which is custom-tailored to effectively process and integrate diverse analytical data acquired from different instruments. GEMTELLIGENCE consists of a combination of strided convolutional neural networks[31] and a variant of the popular Transformer architecture[32]. Specifically, for processing spectral data, we draw inspiration from the work of Ho et al.[33] and use a modified version of their architecture with a bigger kernel size to increase the receptive field in order to capture more global features. The transformer-like component of GEMTELLIGENCE is used to process tabular data and is based on the architecture introduced in ref. 34. The final architecture combines all these elements in a single model enabling end-to-end multi-modal training. As robustness and consistency are key desiderata in gemstone analysis, we include an additional confidence-thresholding scheme in our pipeline allowing users to control the trade-off between the degree of automation provided by GEMTELLIGENCE (i.e. the number of stones that can be processed automatically) and its level of accuracy. As illustrated in Fig. 1c, given a new test stone, we compare the most likely prediction of GEMTELLIGENCE with a threshold value and accept the prediction only if the corresponding probability exceeds this threshold. The value of this threshold is determined by imposing that the trained model attains the desired accuracy level on the training set. More details about the

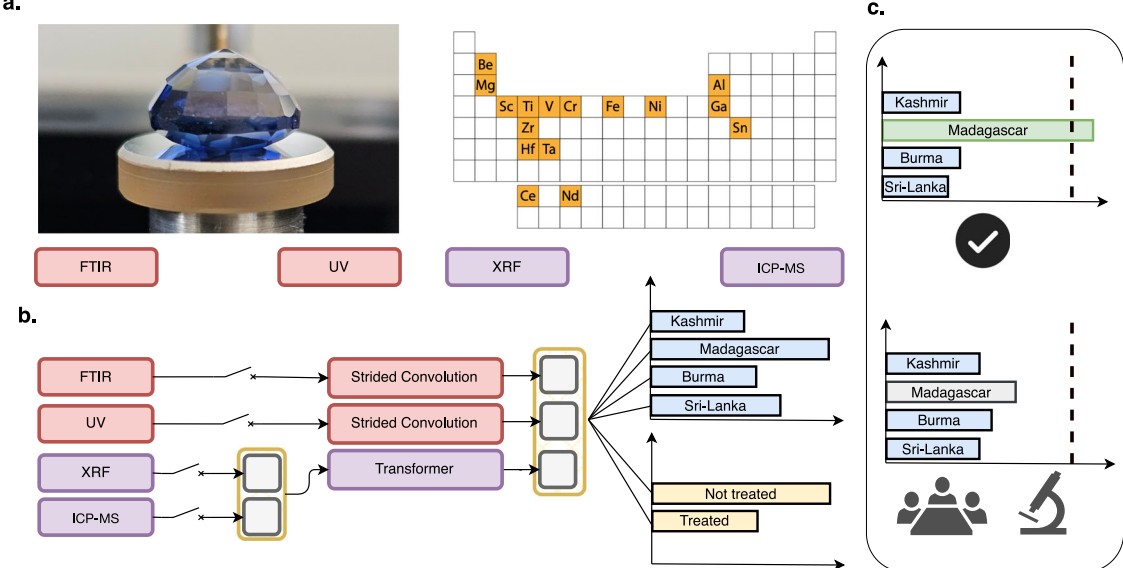

**Fig. 1 | A diagram summarising the workflow of GEMTELLIGENCE.**
**a** GEMTELLIGENCE can process measurements from four distinct data sources: FTIR (Fourier-transform infrared spectroscopy) and UV (ultraviolet-visible-near-infrared spectroscopy) and ICP-MS (laser-ablation inductively-coupled-plasma mass spectrometry) and XRF (energy-dispersive X-ray fluorescence). **b** With a negligible inference time it can predict the probability of a gemstone's origin or whether it has undergone heat treatment. Not all data types are required

for inference; missing sources can be masked out as illustrated by switch symbols in the figure. **c**. Predictions with high confidence (top panel) are accepted, while those that falls below the threshold (bottom panel) should be further analyzed by gemological experts. The value of the threshold, selected during the confidence-thresholding phase, determines the balance between the number of stones that can be processed automatically and the accuracy achieved by the model.

model architecture and the confidence-thresholding procedure can be found in the Methods section.

Our work contributes to the burgeoning field of laboratory automation[35–38], which has seen a rising focus on leveraging Artificial Intelligence (AI) techniques to streamline the time-consuming and repetitive pipelines typical of experimental scientific research. As shown by our empirical evaluation, the implementation of GEMTELLIGENCE in gemological laboratories can assist human experts in the time-consuming task of data assessment and interpretation, allowing them to focus on more high-value activities, including research and development.

## Results

In this section, we present the results of GEMTELLIGENCE for the two challenging tasks of OD and TD of blue sapphires. Blue sapphire is a type of corundum, $Al_2O_3$, with a blue hue caused by the presence of trace amounts of Fe and Ti. Our focus on sapphires stems from two key factors. Firstly, sapphires are widely acknowledged to present more difficulties in achieving accurate OD than other gemstones[11]. Secondly, the TD of sapphires has not been researched as extensively as that of other gemstones like rubies. In the following, we first introduce the tasks, datasets, training, and testing pipelines. Then, we assess GEMTELLIGENCE's performance in processing diverse multi-modal data, through various ablation studies and in comparison to human experts. Additional experiments are also presented in Supplementary Notes 1, 3, 4 and 8.

### Background and experimental setup

**Origin determination.** The problem of OD can be cast as a classification task: based on the laboratory tests performed on a particular gemstone, the goal is to determine its geographical origin out of a discrete set of candidates. Traditionally, this is achieved by comparing the pattern of gemological properties exhibited by the unknown stone to the patterns of properties observed in gemstones from known origins, commonly referred to as reference samples. These reference samples, typically collected by authorized individuals at or near the gemstone's mining site, have precise provenance records confirming their origin. This identification is possible because the geological environment and rock varieties in which blue sapphires develop have specific attributes[39], resulting in slightly varying gemological characteristics[11]. Microscopy and ICP-MS are widely considered the most reliable analytical evaluations for OD. UV and XRF are also employed, though their analysis results are not as reliable as those from microscopy and ICP-MS. In the main body, we focus on the top four major sources of high-value blue sapphires, which make up over 90% of the market's high-quality blue sapphire volume. These sapphires were created through metamorphism and are sourced from metamorphic deposits from Kashmir, Burma/Myanmar, Sri Lanka, and Madagascar. Additional information about the gemstones considered for the study and other experiments with other minor and/or non-metamorphic origins can be found in supplementary note 5 and 6.

**Artificial heat treatment detection.** Artificial heat treatment is a process that involves heating gemstones to enhance their visual appearance, clarity, and color. To determine if a stone has been artificially heat treated, the primary method is to use visual microscopic inspection complemented with spectral measurement techniques. Heat treatment can induce structural changes, especially in microscopic or submicroscopic inclusions. These inclusions, which are solid, liquid, or gaseous phases trapped or formed within the crystal during or after its growth in the earth[40] may become unstable and disintegrate when a gemstone is heated. FTIR and UV analysis are typically employed to support heat treatment detection on sapphires. Since elemental analysis methods such as XRF and ICP-MS cannot capture the underlying physical change of a stone undergoing heat treatment, they are not used for this task. Hence, we also avoid using them as input to our algorithm in order to prevent the introduction of spurious correlations. We frame the TD problem as a binary classification task, comprising a "treated" and a "non-treated" class.

**Training and testing datasets.** The data used for this study comprises over 5500 blue sapphire measurement records obtained from the Gubelin Gem Lab over a seven-year period. For approximately half of the available stones, a comprehensive assessment was performed, resulting in two optical and two chemical spectroscopic measurements (UV, FTIR, XRF, and ICP-MS) being available for each stone. The remaining stones followed a reduced protocol with fewer analyses. Given the limited size of our dataset, we used five-fold cross-validation to obtain a more robust estimate of the model's performance. This involved splitting the data into five subsets, using one for testing and the remaining four for training. As detailed in the next paragraph, we further excluded stones from the test subsets lacking a reliable ground truth estimate. The number of available measurements for each data source across all sets is presented in Supplementary Table 2. Additional details about the stone the devices and methods used to collect the data can be found in the Materials subsection.

**A Note on the ground truth.** Our dataset includes records of high-quality gemstones commonly traded in the market. However, due to the unrestricted movement of these gemstones, the provenance of the vast majority is unclear or nonexistent. For instance, in the case of Kashmiri sapphires, where the main mining activity lasted a few years only and stopped at the end of the 19th century, the number of sapphires with a confirmed provenance is very limited. As a result, for many stones OD and TD cannot be determined with absolute certainty but must be inferred from laboratory analysis results. While this process can potentially introduce inaccuracies, we have relied on the professional OD and TD determinations conducted by the Gübelin Gem Lab. These determinations adhere to industry standards, which encompass a combination of visual, spectroscopic, and chemical methods and properties, and especially when applied to unique inclusions or clear patterns observed in trace-element analysis, offer a high degree of accuracy. In addition to further reduce the likelihood of ground truth errors in the test set, we implemented additional precautionary measures. We only considered stones that meet the following criterias: 1) each measurement is examined independently from the others; 2) all possible relevant measurements are taken; 3) two expert gemologists independently reach the same conclusion through visual inspection for TD; 4) The OD results obtained from ICP-MS align with the findings of visual inspection. Supplementary Note 5 provides details on stone origin and quality.

### Performance evaluation

**AI-supported decision system.** We begin our analysis by comparing the performance of GEMTELLIGENCE with that of human gemologists on various combinations of data sources and tasks. The gemologists follow a strict procedure throughout the process of analyzing the gemstones. First, each data source (e.g. XRF) is observed independently and without access to other information. Second, based solely on this data source, a preliminary conclusion is made regarding the gemstone's origin and any potential heat treatment it may have undergone. We use these sub-conclusions to create statistics and perform comparisons between GEMTELLIGENCE and human experts using individual and combined data sources.

For the sake of fairness, we conduct this comparison by considering only the data sources that were not used to determine the label of the test data. Specifically, we do not use ICP-MS data for OD as the test set was created such that the final conclusion had to match the ICP-MS sub-conclusion, which is regarded as the most reliable data source. Additionally, gemologists examine UV and FTIR spectra simultaneously for TD. Therefore, separate statistics for these two sources are unavailable in the TD case.

Figure 2 shows this comparison between GEMTELLIGENCE and human experts on the OD and TD tasks in terms of the number of stones they can confidently classify based on single or combined data sources and the obtained levels of accuracy.

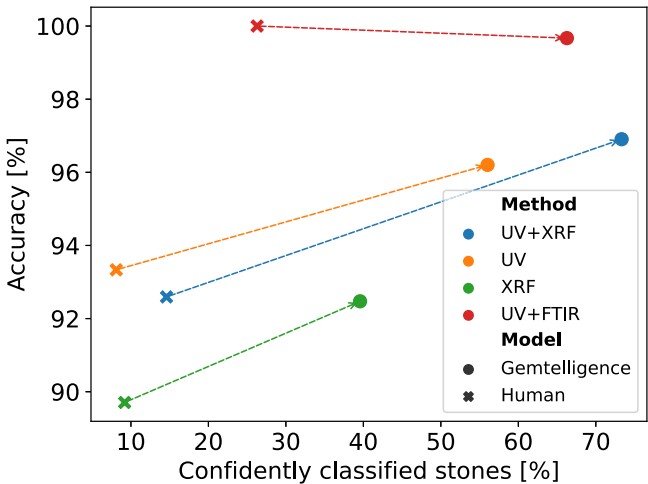

**Fig. 2 | Comparison between human experts and GEMTELLIGENCE (Human and Gemtelligence respectively in the legend) in terms of the size of the subset of stones that have been confidently classified (on the x-axis) and the corresponding level of accuracy achieved for this subset (on the y-axis).** Each color corresponds to a different combination of data sources. All the combinations apart from the red one (UV+FTIR) are used for origin determination while UV+FTIR is used for detection of treatment. The dashed lines are used to highlight the performance change between humans and our model. The results in the plot are obtained by evaluating the performance of experts and GEMTELLIGENCE on test data. In the legend, the various combinations of data sources (UV, XRF, UV+FTIR, and UV+XRS) indicate the data types the model can access at test time. FTIR stands for Fourier-transform infrared spectroscopy, UV for ultraviolet-visible-near-infrared spectroscopy, ICP-MS for laser-ablation inductively-coupled-plasma mass spectrometry and XRF for energy-dispersive X-ray fluorescence.

For GEMTELLIGENCE, we refer to a stone being confidently classified if the probability of the model associated with its final prediction exceeds the threshold value (see Fig. 1c.). In this specific experiment, the threshold has been determined by calibrating the model on the training data to match or surpass the accuracy levels reached by human experts on the test data. On the other hand, we consider a stone confidently classified by human experts if they are certain enough to reach a single, unambiguous conclusion - for example, assigning one country of origin (OD) and one thermal treatment (TD). For all the considered combinations of data sources, GEMTELLIGENCE can provide confident predictions on substantially larger sets of stones (x-axis) than human experts, who are often unable to draw definitive conclusions due to uncertainty. Furthermore, GEMTELLIGENCE also achieves either comparable or higher accuracy levels (y-axis) than human experts while delivering a final conclusion on much larger groups of stones. This experiment demonstrates that, for a substantial portion of gemstones, gemological analysis can be automated with GEMTELLIGENCE while achieving comparable or even higher levels of accuracy. This is noteworthy because in professional gem labs, which need to adhere to best practice benchmarks, the time dedicated for the assessment of the raw analysis is substantial. We proceed with our analysis by exploring how different threshold values yield different trade-offs between accuracy and automation. A higher threshold value results in more stones requiring further human expert analysis but potentially higher accuracy, while a lower threshold value may require less human expert analysis but potentially lower accuracy. Table 1 presents the performance of our model, in three operating setups, namely None, Mode 1 and Mode 2, each offering different trade-offs between automation and accuracy. For the last two modes, the threshold of GEMTELLIGENCE is chosen to ensure a specific accuracy rate (98% for Mode 1 and 99% for Mode 2) on the training stones that have a classification probability above this threshold. For more details on the confidence-thresholding procedure, please refer to Methods section. For the None setup (second and fifth columns in

Table 1), the model is not calibrated since the threshold is set to zero. This means that all predictions are accepted, resulting in the complete automation of the process (100% stones above the threshold) at the cost of higher error rates. However, in scenarios where the model's prediction uncertainty is higher, this configuration lacks preventive measures to mitigate the risk of potential errors. For the other two setups, the threshold is non-zero, resulting in fewer accepted predictions and increased accuracy compared to the None setup. The lower threshold for Mode 1 compared to Mode 2 results in a substantial increase in the number of confidently classified stones for both tasks. Compared to Mode 2, this setup greatly reduces the workload of gemologists, as the inference time of GEMTELLIGENCE is negligible (less than a second), whereas taking a final decision on a single stone can take several hours for human experts. Nevertheless, Mode 1 also leads to a slight reduction in test accuracy, although the results remain favorable and comparable to human capabilities.

In the field of gemstone analysis, the level of automation and accuracy of predictions can substantially impact the workload of gemologists and the value of stones. Depending on the specific use case, it may be advantageous to prioritize one over the other. As incorrect evaluations can impact stone prices, Mode 2 mode represents a more conservative and low-risk configuration as it results in high accuracy levels, despite decreasing the level of automation of the model (number of stones confidently classified). In Supplementary Note 7, we present two practical examples of using GEMTELLIGENCE alongside expert analysis for characterizing two stones of unknown origin.

**Influence of different data sources.** Figure 3 illustrates the relationship between GEMTELLIGENCE's accuracy and the number of confidently predicted stones for OD (left) and TD (right), where a stone is considered confidently classified by GEMTELLIGENCE if the model's probability associated with its final prediction exceeds the threshold value.

The figure reveals a consistent trend across all data sources and tasks: higher levels of confidence in GEMTELLIGENCE's prediction lead to higher accuracy. In other words, when the model assigns a high probability to a certain class, it is more likely to be correct. The observed strong correlation between accuracy and confidence in this experiment validates the effectiveness of our confidence-thresholding procedure.

The results indicate that, for OD, using ICP-MS data leads to ~ 4% higher accuracy compared to the next best single data source, UV, across the entire range of values on the x-axis. This highlights the high-quality information provided by ICP-MS data for this task. Moreover, combining UV and XRF data sources yields a model with comparable performance to that obtained with ICP-MS data, despite the latter's higher complexity and cost. This suggests that combining standard and less expensive analytical data sources can be as effective as the more expensive ICP-MS method.

In TD, while the best results are achieved by integrating UV and FTIR data, GEMTELLIGENCE still reaches similar levels of accuracy when only utilizing FTIR data. This is noteworthy since experts typically rely on both data sources to reach a final conclusion in their analysis and suggests that GEMTELLIGENCE can serve as a valuable tool even in scenarios where using multiple data sources is not feasible.

**Prediction consistency analysis.** To evaluate the accuracy and reliability of GEMTELLIGENCE, we analyze whether the network provides consistent results for the same gemstone when data is collected from different instruments, at various times, and under varying conditions.

Since gemstones often undergo multiple analyses during their lifespan, inconsistent evaluation results can raise doubts about the authenticity of the asset, leading to legal and financial complications. Therefore, assessing whether the predictions remain consistent over time is crucial.

Figure 4 illustrates the predictions of GEMTELLIGENCE for the OD (left columns) and TD (right column). The focus is on gemstones that underwent multiple evaluations over the years, using the None (first row), Mode 1 (second row), and Mode 2 (third row) model setups.

Each stone in our collection that was assessed more than once is represented in each of the six panels of the figure by a line connecting the

**Table 1 | Calibration accuracy used to determine the threshold (first row), the corresponding number of analyzed stones (second row), and test accuracy (third row) for the three considered operating setups both for OD and TD**

| GEMTELLIGENCE setup | Origin Determination | | | Heat Treatment Detection | | |
|---|---|---|---|---|---|---|
| | None | Mode 1 | Mode 2 | None | Mode 1 | Mode 2 |
| Calibration accuracy | None | 98% | 99% | None | 98% | 99% |
| Stones above threshold | 100.0% | 74.2% | 38.5% | 100.0% | 97.4% | 95.5% |
| Test accuracy | 90.69% | 96.8% | 99.1% | 98.03% | 98.7% | 98.9% |

OD was performed using UV and XRF and TD was done using UV and FTIR. Refer to Supplementary Note 2 for details on the performance of the model for each specific class.

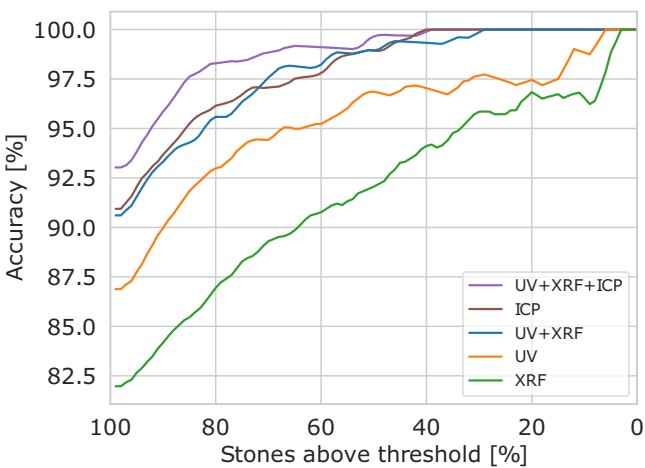
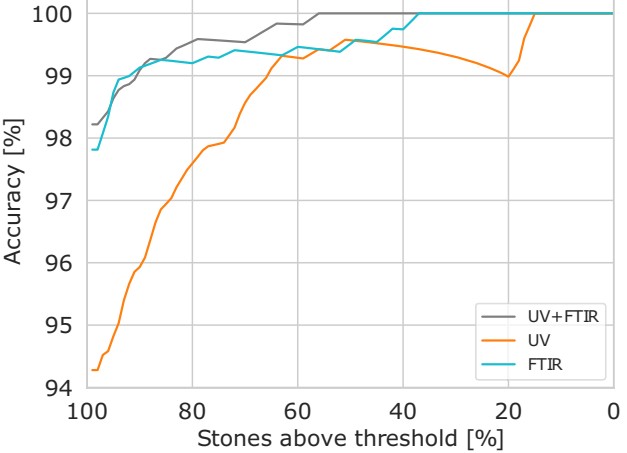

**Fig. 3 | Accuracy (%) vs. stones above the threshold (%).** The left subfigure shows results for origin determination and the right subfigure for detection of treatment, with different data sources provided as inputs to the model. The x-axis represents the number of stones confidently classified by GEMTELLIGENCE with a probability greater than a fixed threshold value (see Fig. 1c). Starting from the left and moving towards the right on the same axis, we gradually increase the threshold and indicate the number of stones that the model confidently classifies for each resulting subset, along with the corresponding accuracy (y-axis). Data sources not present in the legend are masked. In the legend, the various combinations of data sources (UV +XRF+ICP, ICP, etc) indicate the data types the model can access at test time. FTIR stands for Fourier-transform infrared spectroscopy, UV for ultraviolet-visible- near-infrared spectroscopy, ICP-MS for laser-ablation inductively-coupled-plasma mass spectrometry and XRF for energy-dispersive X-ray fluorescence.

different evaluations across time. A black line is drawn if the model's predictions are consistent across evaluations, while a red line is drawn if they are not. Dots located at the extremes of the line indicate the predictions of GEMTELLIGENCE: the absence of a dot implies that GEMTELLIGENCE's confidence did not exceed the threshold and hence, a decisive conclusion could not be drawn. It is worth emphasizing that scenarios, where uncertainty prevents a change in prediction (indicated by black lines and no dots), are generally more desirable than inconsistent predictions (shown in red lines). Typically, uncertainty prompts experts to carry out additional analyses to reduce the chances of making mistakes.

The results displayed in Fig. 4 demonstrate that even without a threshold, i.e. None setup, (upper row), GEMTELLIGENCE exhibits a good level of consistency in its predictions, with only 23 inconsistent predictions out of 148 for OD and zero out of 62 for TD. However, when GEMTELLIGENCE is employed in Mode 1 and Mode 2 setups (middle and bottom rows), all inconsistent predictions disappear as the model's outputs for such cases fall below the threshold, reflecting its uncertainty for those particularly challenging and ambiguous samples.

This experiment highlights the value of our confidence-thresholding methodology, which enables the user to disregard predictions that the model is not highly confident about, reducing the risk of incorrect predictions.

## Conclusion
### Discussion
This study presents GEMTELLIGENCE, a multi-modal deep learning approach for automated origin determination and heat treatment detection of gemstones. GEMTELLIGENCE demonstrates exceptional capabilities in handling and combining complex and varied data structures, leveraging correlations between different data modalities to enhance prediction accuracy. Its architecture, incorporating transformers and convolutional neural networks, enables flexible gemstone classification using various combinations of data sources and allows simultaneous end-to-end processing of tabular and spectral data. The introduction of GEMTELLIGENCE offers several notable advantages. Firstly, its outputs are well-calibrated as it provides correct predictions with high confidence on a large percentage of test samples. This contrasts with expert-based evaluations that provide confident predictions for a much smaller subset of stones. Secondly, GEMTELLIGENCE achieves excellent results by leveraging inexpensive data sources, thereby reducing reliance on costlier analytic methods such as ICP-MS.

Overall, GEMTELLIGENCE has the potential to drastically benefit the gemstone industry. Its implementation offers substantial cost savings and allows human experts to focus on more valuable activities in research and development. By deploying GEMTELLIGENCE, the gemstone analysis process can be standardized, substantially reducing the incidence of ambiguities and increasing trust levels in the entire marketplace. In conclusion, we anticipate that our results, accompanied by publicly available code and data, will stimulate more investigations in this domain and advance the development of techniques and tools for gemstone analysis automation. Moreover, given the extensive utilization of the spectroscopic analysis tools examined in this study, we envision that exploring the potential application of the GEMTELLIGENCE framework beyond the field of gemology would be a captivating area for future research in material science.

**Fig. 4 | GEMTELLIGENCE predictions over time.**
The subfigures on the left show results for country of origin (OD) and the subfigures on the right for detection of treatment (TD). Each horizontal line is used to indicate a different stone with a unique idetifier (Stone ID) that is analyzed multiple times over the years. The top row is with no threshold, the middle row with `Mode 1`, and bottom row with `Mode 2` setup.

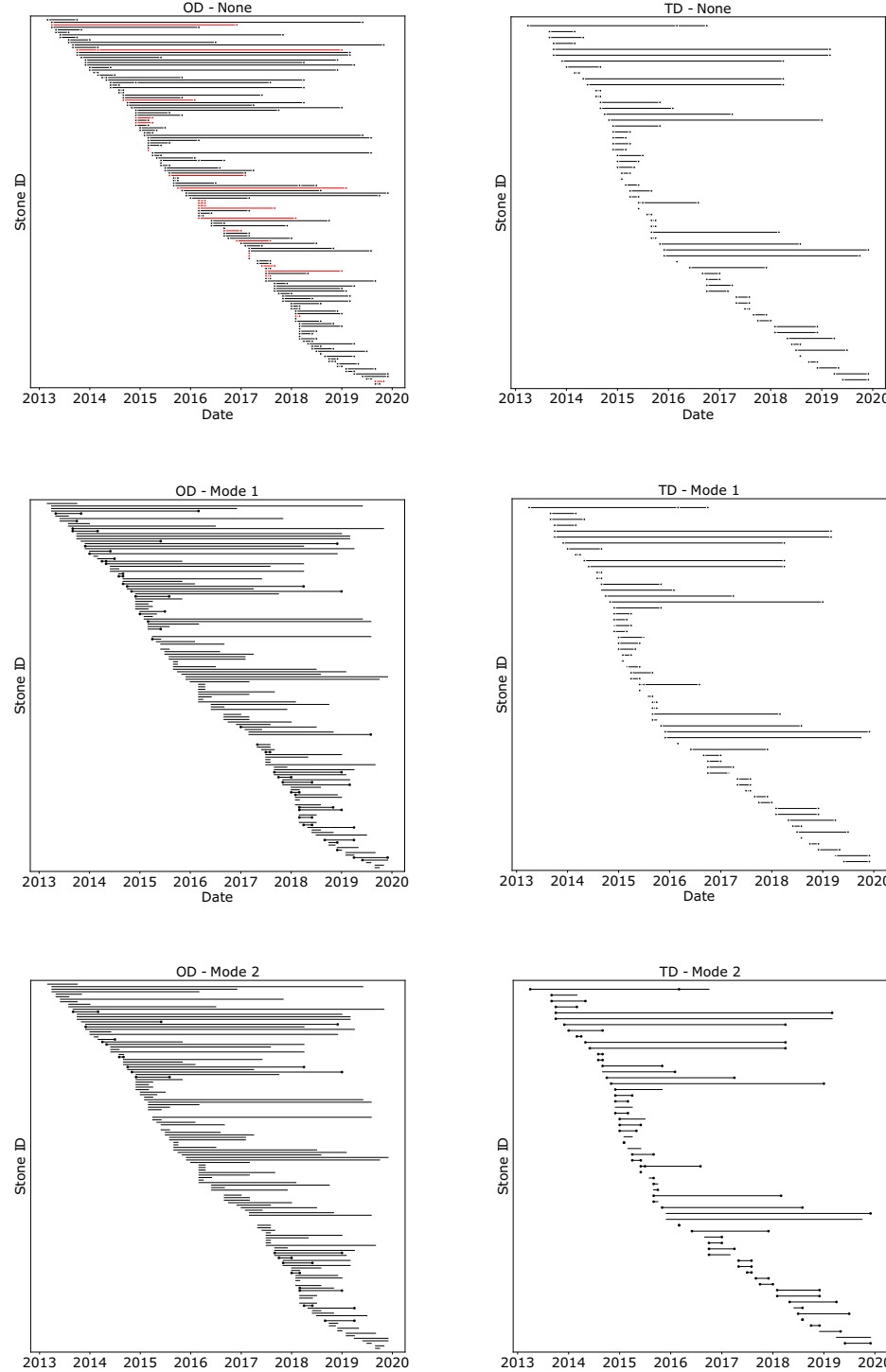

## Limitations

GEMTELLIGENCE, in its current version, has a number of limitations. First, GEMTELLIGENCE is trained exclusively on metamorphic stones, which despite covering a large part of market value stones, do not represent the entirety of available stone classes. This limitation mainly arises from the lack of abundant labelled datasets on non-metamorphic gems. To take this aspect into account, our analysis in Supplementary Note 6 shows that a simple pre-classifier can be used to discriminate between metamorphic and non-metamorphic stones and hence, select the stones that the network can classify meaningfully (metamorphic) and those that require human analysis (non-metamorphic). While our experiments show that this approach is

effective, it would be desirable for GEMTELLIGENCE to process any kind of input gemstone without the necessity for a pre-processing step. A second limitation arises from the experimental evaluation. GEMTELLIGENCE TD and OD prediction ability is validated with respect to determinations conducted at the Gübelin Gem Lab. While these are highly accurate, they may not always perfectly reflect the true origin of a gemstone, as detailed in the "A Note on the Ground Truth" paragraph. While we do not expect this to be a strong limitation, as the procedure by which labels are created follows stringent professional standards, the performance and reliability of the model could eventually benefit from reducing the noise of the labels or any potential bias. Both limitations can be addressed by the introduction of

refined datasets including a larger set of gemstones analysed from different gemstones laboratories or with clear provenance. If such datasets are available, GEMTELLIGENCE can be easily retrained and eventually improved. We leave these research avenues to future work.

## Methods
### Materials
The stones used in this study for training and testing are high-quality metamorphic blue sapphires tested by the Gübelin Gem Lab between 2013 and 2020. Metamorphic sapphire is formed in alumina-rich rocks under amphibolite and granulite facies conditions, at temperatures between 500° and 800 °C[41]. Gem-quality metamorphic sapphire is typically found in secondary (placer) deposits originating from complexes of ultramafic host rocks, marble, quartzite, gneiss or metapelite, intersect by plutonic rocks[42]. The particular formation process is governed by isochemical or metasomatic metamorphism affecting the different lithologies. The age of formation of the sapphires from Kashmir and Burma/Myanmar is related to the Cenozoic Himalayan orogeny that occurred 45-5 Ma ago, while the much older Pan-African orogeny caused the formation of the sapphires from Sri Lanka and Madagascar, some 750-450 Ma ago[12]. The origin of the stones were determined by comparing their data (visual, chemical and spectroscopic) with the data of reference samples, i.e. stones from a known origin. Direct radiometric age determination using the U-Pb method on syngenetic zircon inclusions trapped in sapphires was applied on a small number of the stones used in this study, providing additional strong evidence to separate the sapphires from the Himalayan and the Cenozoic orogenies[43], and positively corroborating the ground truth.

### Data sources
The following devices and methods were utilized to collect the data used in this study:

**ICP-MS.** For ICP-MS data, we used an Elemental Scientific (ESI) 193 nm excimer laser ablation system with a large-format sample chamber and a small-volume, flexible cup that collected the ablated material. Three ablations were created for each stone, having a 50-micrometer diameter spot size, 15 Hz repetition rate, and 6 J/cm$^2$ fluence. For each ablation, the materials were conveyed to the ICP via a blend of He (1000 ml/min) and Ar (700 ml/min) gases, where the material got ionized. Finally the ions were transported to an Agilent 8800 mass spectrometer, which measured the following elements / isotopes: $^7$Li, $^9$Be, $^{25}$Mg, $^{27}$Al, $^{29}$Si, $^{45}$Sc, $^{47}$Ti, $^{49}$Ti, $^{51}$V, $^{52}$Cr, $^{53}$Cr, $^{55}$Mn, $^{56}$Fe, $^{57}$Fe, $^{59}$Co, $^{62}$Ni, $^{71}$Ga, $^{89}$Y, $^{90}$Zr, $^{93}$Nb, $^{118}$Sn, $^{140}$Ce, $^{146}$Nd, $^{176}$Hf, $^{181}$Ta, $^{193}$Ir, $^{195}$Pt. The acquired data was then processed from counts/second to concentration using Glitter[44], with NIST 612[45] as the primary calibration standard and BHVO-2G[46] and ATHO-G[47] as secondary standards. A value of 99 wt% $Al_2O_3$ was used as an internal standard for all corundum. Following gemological-driven analysis, we focussed for our study on the following entries: $^9$Be, $^{25}$Mg, $^{27}$Al, $^{45}$Sc, $^{49}$Ti, $^{51}$V, $^{53}$Cr, $^{57}$Fe, $^{62}$Ni, $^{71}$Ga, $^{90}$Zr, $^{118}$Sn, $^{140}$Ce, $^{146}$Nd, $^{176}$Hf, $^{181}$Ta.

**FTIR.** Non-polarized FTIR spectra were collected in air using a Varian 640 FTIR spectrometer equipped with a KBr beam splitter and a deuterated triglycene sulfate (DTGS) detector. For each sample, three measurements in perpendicular directions were conducted either using diffuse reflectance (DRIFT) or with transmitted light. For each measurement, a total of 64 scans with a resolution of 1 cm$^{-1}$ to 4 cm$^{-1}$ were collected and averaged. This was done for the wavenumber range of 200 cm$^{-1}$ to 7000 cm$^{-1}$, with a background collected at regular intervals. As the measurements had different intervals and offsets due to differences in software version and settings, we homogenized the data so that every spectrum had a step size of 1 cm$^{-1}$. This was done by a cubic spline interpolation on the available data. As not all data were collected over the range of 200 cm$^{-1}$ to 7000 cm$^{-1}$, we padded the missing values with zeroes. Further, any spectra that had a measurement with a value smaller than -5 or greater than 10 were dropped as these values were extreme

outliers and not in the expected range of the measurement. This filtering reduced the data set by less than 1%. The resulting spectral data consisted of 6801 data points per measurement.

**XRF.** XRF (ED-XRF) measurements of major, minor, and trace elements were conducted using a Thermo Fisher Scientific QUANTX, with a silicon drift detector (SDD), 1 mm collimator, and over an applied energy range of 4-50 kV, with a variety of filters used to reduce spectral interferences on critical elements. For blue sapphires, the only minor and trace elements consistently detectable are Ti, Cr, V, Fe, and Ga, along with the major element Al. In order to identify treatments or synthetic samples, Pb, W, and Pt were included during most blue sapphire measurement routines. For blue sapphires, we discarded the ED-XRF measurement, if
- The $Fe_2O_3$ value is above 40'000 ppm
- The $Al_2O_3$ value is under 850'000 ppm
- The $Cr_2O_3$ value is above 10'000 ppm
- The $TiO_2$ value is above 6'000 ppm

Such outliers do occur occasionally in XRF measurements due to various reasons such as diffraction peaks induced by the crystal structure of the minerals. ED-XRF data is tabular in nature, having 26 entries describing the concentration of certain chemical compounds.

**UV.** Polarised UV (UV-Vis-NIR) spectra were collected using a Varian (now Agilent) Cary 5000, using deuterium and tungsten halogen light sources and indium gallium arsenide (InGaAs) detector. Measurements were performed over the wavelength range 280-880 nm with a step size of 0.5 nm, using both a reference and sample line equipped with polarisers and beam condensers. In most cases, two measurements in perpendicular polarisations were taken on each sample. In the case of single measurements, the measurement was duplicated to be consistent with the bi-polar measurements. As the absorbance can not be negative, any spectra with negative values were discarded. This could occur due to faulty measurements. The resulting final data sample consisted of 2 x 1201 entries.

During the time period from which the data included in this study were obtained, several variants of these instruments were used and a minority of data were collected on other instruments, not detailed here. Data consistency between these different models was maintained through standardized acquisition protocols and the use of identical calibration and secondary reference materials.

### GEMTELLIGENCE architecture
GEMTELLIGENCE is an artificial neural network created to process multi-modal data from gemological laboratories. It is composed of a UV encoder, an FTIR encoder, and a single elemental analysis encoder that processes XRF (and optionally ICP-MS jointly) data. The encoders generate embeddings which are then combined by the network's head. This head comprises a concatenation layer to combine the encoders' outputs, batch normalization, and a final linear classification layer.

The UV and FTIR encoders are strided convolutional neural networks with skip connections as proposed in[33]. At the core of their architecture, there are six residual connection layers, each with a hidden dimension of 128, kernel size of 17, and strides of 2. These blocks are preceded by a first convolution layer of kernel size 59. For UV measurements, which involve two spectra taken in perpendicular directions, the input channel of the first convolution layer has a dimension of two, while for FTIR measurements, a single spectrum is used and the first convolution has a dimension of one. After the skip connection blocks, the FTIR and UV encoders employ a single convolution channel mapping the hidden dimension from 128 to 1, resulting in final embeddings of length 213 and 190 respectively. The parameter selection was based on a preliminary grid search. Particularly, we found that a smaller kernel size or fewer residual connection blocks caused a decrease in performance, while larger dimensions resulted in high memory usage and slow training with no increase in accuracy. The elemental analysis encoder is based on the `SAINT` framework introduced by[34], which was

specifically designed to provide a sample-efficient deep learning method for tabular data. We opted to follow the `Both` configuration of the original paper, which deploys both intrasample and intersample attention mechanisms. Intrasample attention is a standard self-attention mechanism, operating on input features (rows), while intersample attention compares specific input features across different samples (columns). Our implementation follows the same hyper-parameters as described in the original paper. However, since our setting only has one sample at inference time rather than a batch, we decided to pre-append a series of reference stones to the batch for both training and testing to avoid a shift in the distribution caused by the intrasample mechanism. XRF and ICP-MS data are concatenated before the encoder allowing for the model to learn dependencies between both data types. The output tensor of the elemental analysis encoder has a single hidden dimension with a length of 32.

Finally, the head of the network is composed of a concatenation layer, followed by a batch normalization and a readout layer. The concatenation layer combines the one-dimensional embeddings from the UV, FTIR, and elemental analysis encoders into a single tensor by concatenating them along the time dimension. This tensor is then fed into the batch normalization and classification layers. The readout layer is composed of a linear layer with a softmax activation function, and its output is the final classification probability for each class.

### Training and testing GEMTELLIGENCE

**Experimental setup**. For training our model, we randomly partitioned the training data into 80% for training, and 20% for validation, saving the model's weights every 5 epochs during the 250 epochs. We then picked the best model in terms of accuracy from the saved weights. During training the batch size was set to 16 and the learning rate was set to 0.0001, with an automatic decay of factor 10 if there was no improvement for more than 10 epochs. To allow the model to learn to handle missing data, we randomly masked each data source with a probability of 0.5 during training, replacing the values with the mean value across the dataset. We did not perform any data normalization or augmentation, as it was found to be detrimental in early experiments. We repeated the same procedure for each of the five folds in the cross-validation procedure. We evaluate the performance of both OD and TD models using five-fold cross-validation. The main advantage of this procedure is that it provides a more reliable estimate of the model's performance than a simple train/test split, especially when the dataset is relatively small, as in our case. The data are randomly split into five sets, of which one is used for testing and the remaining four are used for training. This process is repeated for each fold. The training data are used to train and calibrate the model, while the test data are employed to measure the model's performance. In each fold, the test data was not used, neither for training nor validation.

In order to generate the final results from the folds, we followed the procedures laid out in ref. 48. Specifically, for all results apart from Fig. 3, we concatenated the predictions of GEMTELLIGENCE from each fold and then calculated the final statistic. For Fig. 3 we first computed the curve in each fold, and then obtained the final curve by computing the average of the curves from each fold. We conducted all the experiments on a machine equipped with an NVIDIA GeForce RTX 2080 Ti with 12 CPU cores.

**Confidence-thresholding procedure**. The purpose of our confidence-thresholding procedure is to determine the reliability of the model's prediction based on the associated level of confidence. More formally, we define the model's confidence $c$ for a given prediction $p$ as the maximum value of the last softmax layer. A reliable prediction is defined as a prediction for which $c$ is greater than some predefined threshold $\hat{c}$ (e.g., 0.95). To determine the value of $\hat{c}$, we perform the following steps: first, we compute the model's predictions $\{p_i\}_{i=1}^{N}$ and associated confidence values $\{c_i\}_{i=1}^{N}$ for each stone in the training set after training. Then, we sort the stones by confidence values from lowest to highest (i.e. $c_{(1)} \leq c_{(2)} \leq \ldots \leq c_{(N)}$). Next, we iteratively compute the accuracy of the subset of stones with the least confidence

removed until the subset accuracy is greater than a pre-specified value $\epsilon$ (e.g. 95%). We define the accuracy of the subset of stones corresponding to the entire dataset minus the $k$ stones with the smallest confidence values as $\mathcal{A}_{N-k}$. At inference time, the threshold of the least confident stone in this subset is used to decide if a model prediction is reliable. Specifically, we set $\hat{c} = c_{k^*}$ where where $k^*$ is defined as the index that satisfies $\mathcal{A}_{N-k'} \geq \epsilon$ for the first time, where $k' \in 1, 2, \ldots, N$.

## Data availability

A portion of the dataset is publicly available following the instructions on the project's GitHub repository[49]. Researchers can request access to the entire dataset upon reasonable request.

## Code availability

The source code for GEMTELLIGENCE is available on the project's GitHub repository[49].

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

## Acknowledgements
We especially thank Klemens Link and the entire CSEM team for their insightful preliminary discussions and ideas. We acknowledge Innosuisse project 41844.1 IP-ICT for the funding.

## Author contributions
T.B. and L.B. designed the experiments and T.B. implemented the code. D.N. and A.G. collected and labeled the data. The structure of the manuscript was designed by T.B. and L.B., and the paper was mainly written by T.B. and L.B., with substantial feedback provided by D.N., A.G., O.F., P.T., and M.K. Regarding the project, T.B, L.B., M.K. and O.F. were responsible for the Machine Learning side, while D.N., A.G., and P.T. dealt with the Gemology part. Supervision and coordination of the development of the method was provided by D.N. and M.K., and the design and writing of the manuscript was supervised by D.N. and O.F. T.B. is the corresponding author.

## Competing interests
The authors declare no competing interests.

## Ethics
Three authors are employed at Gübelin Gem Lab Ltd, a for-profit company that makes commercial use of the GEMTELLIGENCE.
