## [Peer Review File · Communications Engineering]

Reviewers' comments:

Reviewer #1 (Remarks to the Author):

Deep learning shows strong potential for applications in processing large amounts of data and image recognition, and is an effective tool for performing gemstone identification in the future. This research creatively proposes a new gemstone identification system GEMTELLIGENCE that can quickly and accurately identify the origin of jewelry and determine the heat treatment. It saves the cost of testing and labor for traditional identification. There have been many attempts to apply machine learning and deep learning to gemstones in the past, the one breakthrough in this research is that the data used to train the model is not limited to tabular data, but also includes image data. But there are several issues that the authors need to take into consideration.

1. Using multiple types of data for deep learning, how to ensure that the model can maintain a balance between different data types instead of being biased to types with large amounts of data, such as images.
2. An application case may give the reader a deeper understanding of how GEMTELLIGENCE works.
3. More background information on the data is needed, such as where the stones were harvested from and the type of host rock, so that the reader has a good understanding of the data.
4. You should give an explanation of the results of deep learning from a gemological perspective, not just a mathematical approach. You can make a feature importance analysis, or something else. For example, for sapphire, you can give a gemological interpretation of the results in terms of predictions for different origins.

Reviewer #2 (Remarks to the Author):

In this paper, the author proposes a deep learning-assisted gemstone origin determination method for distinguishing blue sapphires from Kashmir, Madagascar, Burma, and Sri-Lanka, as well as heat treatment identification. The primary aim is to introduce automation in data analysis, ultimately reducing both time and equipment costs associated with decision-making in gemological laboratories.

The author asserts that the trained AI model can confidently determine gemstone origin without relying on trace element analysis, such as laser-ablation inductively-coupled plasma mass spectrometry (ICP-MS), by solely utilizing information from infrared absorption (FTIR), UV-VIS absorption (UV-VIS), and x-ray fluorescence (XRF). Furthermore, for heat treatment identification, the AI model can achieve similar accuracy using only UV and FTIR data. The AI model significantly accelerates data analysis, reducing the time required from several hours to mere seconds and eliminating the need for expensive ICP-MS analysis in origin determination.

In my opinion, origin determination requires more robust scientific evidence to support the author's claims. Presently, even the most up-to-date research struggles to confidently pinpoint the origin of a gemstone based solely on a combination of spectroscopy, trace-element, and microscopy analyses. It's important to note that the proposed AI model merely serves as a replacement for the current identification criteria employed by one gemological laboratory and does not necessarily lead to more accurate determinations.

As a result, I do not recommend publishing this paper in Communications Engineering.

Overall, the proposed method demonstrates its ability to successfully differentiate between blue sapphires from Kashmir, Madagascar, Burma, and Sri-Lanka and detect heat treatment. However, concerns arise regarding the ground truth of gemstone origin in this paper, which relies on laboratory analysis rather than confirmation from the mine or the original source. While the AI model may replace existing origin determination criteria used by the Gubelin lab, it may lack sufficient scientific evidence for origin determination. Even with the latest trace element analysis, the distribution of trace elements among different origins still heavily overlaps[1]. The author mentions that opinions on origin remain subjective, and using biased opinions as a training model may lead to biased outcomes. The author should revise their claim to emphasize that the AI model primarily streamlines the time-consuming analysis protocol used by the Gubelin lab, rather than overemphasizing its accuracy or reliability due to the questionable training ground truth. Additionally, crucial details about how the ground truth, initial opinions of origin, and heat treatment were determined are missing, such as the utilization of spectroscopy and trace element analysis features. Consequently, there is a lack of scientific evidence supporting the proper generation of the ground truth.

Secondly, it appears that the AI model only considers blue sapphires from Kashmir, Madagascar, Burma, and Sri-Lanka. While these four locations do produce the majority of high-quality blue sapphires, other regions like Montana, Thailand, Tanzania, and Columbia also yield blue sapphires. The author should explain how the AI model performs when evaluating samples from origins beyond the main four. Misidentifying gemstones from these regions could result in significant errors and damage the laboratory's reputation. This paper may oversimplify the challenges faced by gemological laboratories in the field.

Thirdly, the author seems to exaggerate the time required for origin identification. In straightforward cases, like an obviously unheated Sri Lanka sapphire with characteristic inclusions, the process may take only a few minutes when examined under a microscope, without the need for advanced testing like FTIR, UV-VIS, or ICP-MS. Several hours of analysis may only be necessary in situations where characteristic inclusions are absent, and additional time is consumed by gemologists seeking multiple opinions, including spectroscopy analysis.

Finally, while using deep learning for heat treatment detection can reduce reliance on human interpretation and save time, its value compared to spectral analysis algorithms, especially given existing criteria based on the presence of peaks and absorption bands[2,3], should be carefully considered.

References:

1. Aaron C. Palke, Sudarat Saeseaw, Nathan D. Renfro, Ziyin Sun, and Shane F. McClure, "Geographic Origin Determination of Blue Sapphire," *GEMS & GEMOLOGY*, Vol. 55, No. 4, pp. 536–579 (2019).
2. Yuyang Zhang and Meihua Chen, "Identification of Heat-Treated Sapphires from Sri Lanka: Evidence from Three-Dimensional Fluorescence Spectroscopy," *Crystals*, 12(2), 293 (2022).
3. <https://www.gia.edu/doc/Observations-on-the-heat-treatment-of-basalt-related-blue->

sapphires_final.pdf

Reviewer #3 (Remarks to the Author):

In this manuscript, "Gemtelligence," a novel automation model has been proposed to evaluate the origin and heat treatment analysis of blue sapphire. This is due to the limitations of traditional analysis, which requires experienced gemologists and multiple analytical techniques to reach the final decision. The main idea of this model is compelling and tends to be efficiently applied for origin determination and heat treatment analysis. In addition, it is a topic that is gaining attention in the gem industry, and recently, a similar model was introduced for rubies. However, my only concern is that the same article has been published on the author's profile at <https://www.researchgate.net/>, which everyone can easily access via the link https://www.researchgate.net/publication/371490185_Gemtelligence_Accelerating_Gemstone_classification_with_Deep_Learning. Is this considered self-plagiarism? Could the authors explain a difference between the already-published article and this submitted article?

Author Responses

January 2024

General comment

We would like to thank all the reviewers and the editor for taking the time to read the manuscript. We are very glad to hear that all the reviewers recognized the potential of our approach. We are grateful for the questions we received and we believe that, by successfully addressing them and modifying the paper accordingly, the manuscript is now significantly improved. In the following, we address the concerns of each reviewer.

Reviewer 1

Deep learning shows strong potential for applications in processing large amounts of data and image recognition, and is an effective tool for performing gemstone identification in the future. This research creatively proposes a new gemstone identification system GEMTELLIGENCE that can quickly and accurately identify the origin of jewelry and determine the heat treatment. It saves the cost of testing and labor for traditional identification. There have been many attempts to apply machine learning and deep learning to gemstones in the past, the one breakthrough in this research is that the data used to train the model is not limited to tabular data, but also includes image data. But there are several issues that the authors need to take into consideration.

Re: We would like to thank the reviewer for taking the time to read our manuscript. We appreciate that you find our work creative and that you recognize the ability of Gemtelligence to process multiple modalities, beyond only tabular data. Below, we aim to address your raised questions and concerns about the paper:

1. Using multiple types of data for deep learning, how to ensure that the model can maintain a balance between different data types instead of being biased to types with large amounts of data, such as images.

Re: Thank you for raising this point. To prevent the model from becoming biased towards more abundant or informative data types, we employed random masking during training. Particularly, each data source in a training sample was randomly masked with a probability of 0.5. In a preliminary ablation study, we

found this technique to be crucial for training a model that can robustly work even when specific data sources are unavailable. We would also like to clarify that our model does not process image data. Instead, Gemtelligence processes two data types: 1) spectral data obtained from UV and FTIR spectroscopy measurements and 2) tabular data acquired from ED-XRF and ICP analyses. Supplementary Table 2, shows the specific amount of data available for training from the different types. While integrating an additional image encoder to Gemtelligence would have been straightforward and interesting from a scientific perspective, we omitted this study due to the lack of good quality image data from the stones.

2. An application case may give the reader a deeper understanding of how GEMTELLIGENCE works.

Re: Thank you for providing this valuable suggestion. We have developed a web app demo that allows readers to interact with the model’s decision-making process and gain an intuitive understanding of its capabilities. It is included with our submission material, and we would like to invite the reviewer to explore it. This demo together and the entire code base, will also be released open source after the review process.

3. More background information on the data is needed, such as where the stones were harvested from and the type of host rock, so that the reader has a good understanding of the data.

Re: We thank the reviewer for this suggestion. Consequently, we have added a more comprehensive explanation of the stones’ harvesting locations, the types of host rock, and additional details of the identification process. These details can now be found in the “Materials” section and Supplementary Note 6.

4. You should give an explanation of the results of deep learning from a gemological perspective, not just a mathematical approach. You can make a feature importance analysis, or something else. For example, for sapphire, you can give a gemological interpretation of the results in terms of predictions for different origins.

Re: Thank you for your valuable advice. Inspired by your suggestion, we have decided to incorporate a new section titled “Comparison of Gemtelligence and Analytical Methods for Heat Treatment” into the manuscript. In this section, we assess whether well-established FTIR peaks in the gemological literature used for Heat Treatment Detection are also important for Gemtelligence via a counterfactual explanation approach. Our finding shows that Gemtelligence is particularly sensitive to these well established FTIR peaks, such as 3309 and 3232 cm^{-1} . We have chosen to focus on FTIR and heat treatment due to the analysis’s relative simplicity. Key FTIR bands are easily recognizable and the binary classification nature of Heat Treatment Detection makes our analysis easy to understand. However, for readers with more gemological expertise, we

have released the counterfactual explanation tool used for the analysis in our codebase, which works with both UV and FTIR data. We believe this tool will further assist interested readers in understanding Gemtelligence’s decision-making process.

Reviewer 2

In this paper, the author proposes a deep learning-assisted gemstone origin determination method for distinguishing blue sapphires from Kashmir, Madagascar, Burma, and Sri-Lanka, as well as heat treatment identification. The primary aim is to introduce automation in data analysis, ultimately reducing both time and equipment costs associated with decision-making in gemological laboratories. The author asserts that the trained AI model can confidently determine gemstone origin without relying on trace element analysis, such as laser-ablation inductively-coupled plasma mass spectrometry (ICP-MS), by solely utilizing information from infrared absorption (FTIR), UV-VIS absorption (UV-VIS), and x-ray fluorescence (XRF). Furthermore, for heat treatment identification, the AI model can achieve similar accuracy using only UV and FTIR data. The AI model significantly accelerates data analysis, reducing the time required from several hours to mere seconds and eliminating the need for expensive ICP-MS analysis in origin determination.

Re: We would like to thank the reviewer for taking the time to read our manuscript. We are happy that the reviewer found that our work can be useful to the gemological community.

In my opinion, origin determination requires more robust scientific evidence to support the author’s claims. Presently, even the most up-to-date research struggles to confidently pinpoint the origin of a gemstone based solely on a combination of spectroscopy, trace-element, and microscopy analyses. It’s important to note that the proposed AI model merely serves as a replacement for the current identification criteria employed by one gemological laboratory and does not necessarily lead to more accurate determinations. As a result, I do not recommend publishing this paper in Communications Engineering.

Re: We recognize the reviewer’s concern about the inherent complexity of origin determination and agree that it is not always possible to definitively assign an origin to all stones. However, it is important to note that for metamorphic sapphires, there exist characteristic inclusions that strongly support a certain origin, especially for Kashmir, Burma, and Sri-Lanka. Moreover, trace-elemental methods offers high degree of accuracy. Therefore, origin assignment is not feasible only for a small subset of metamorphic stones, such as those lacking characteristic inclusions or for which trace-elemental analyses yield conflicting or inconclusive results.

In our work, as described in the “Training and Testing Datasets” section, for testing the algorithm on OD we utilized only those stones where the sub-conclusions taken by two human experts independently coincided with each

other and with the origin determined by ICP-MS. Consequently, for these stones we have very high confidence about the correctness of the ground truth. For these stones we showed that in fact spectroscopy-based methods are sufficient to achieve comparable results to ICP-MS and microscopy.

We also acknowledge the reviewer’s observation that the stones were analyzed from a single laboratory organization. However, the OD process is based on a well-established method in which stones are compared against reference stones, which have clear provenance, making it independent of the laboratory. Moreover, the methodology we propose is not limited to data solely collected from our laboratory, and it can be applied to datasets from different labs.

Overall, the proposed method demonstrates its ability to successfully differentiate between blue sapphires from Kashmir, Madagascar, Burma, and Sri-Lanka and detect heat treatment. However, concerns arise regarding the ground truth of gemstone origin in this paper, which relies on laboratory analysis rather than confirmation from the mine or the original source. While the AI model may replace existing origin determination criteria used by the Gubelin lab, it may lack sufficient scientific evidence for origin determination. Even with the latest trace element analysis, the distribution of trace elements among different origins still heavily overlaps.

Re: We thank the reviewer for raising this point. As explained in the previous response, we took stringent measures to ensure the reliability of our ground truth data. For TD, we only included stones where two independent experts reached the same conclusion regarding treatment based on visual inspection. For OD, we further required that ICP-MS also confirmed this conclusion. Since both methods rely on comparisons with reference stones and widely accepted criteria, our test set consists of stones with a minimal risk of unreliable labels. We would also like to emphasize that the inherent uncertainty of the ground truth labels is not exclusive to our study and is also present in other scientific fields such as medical diagnosis <https://www.nature.com/articles/d41586-020-00847-2>, or even within the discipline of gemology itself, e.g. color authenticity for green diamonds. Furthermore, a slight difference between the assumed and effective ground truth does not invalidate the achievement of this work, which is to demonstrate that Gemtelligence can achieve predictive performance comparable to ICP-MS and visual inspection by using only spectral methods.

The author mentions that opinions on origin remain subjective, and using biased opinions as a training model may lead to biased outcomes. The author should revise their claim to emphasize that the AI model primarily streamlines the time-consuming analysis protocol used by the Gubelin lab, rather than overemphasizing its accuracy or reliability due to the questionable training ground truth.

Re: We are grateful for the reviewer’s valuable suggestion. This paper presents compelling evidence that Gemtelligence outperforms human experts (see Figure 2) when analyzing gemstones using UV, FTIR, and XRF spectral data. The ground truth for these analyses was established using ICP and Microscopy. We have further refined the abstract and main body of the paper to eliminate any

ambiguity regarding this point and to enhance clarity for readers.

Additionally, crucial details about how the ground truth, initial opinions of origin, and heat treatment were determined are missing, such as the utilization of spectroscopy and trace element analysis features. Consequently, there is a lack of scientific evidence supporting the proper generation of the ground truth.

Re: The Gübelin Gem Lab solely applies well-established and published methods and criteria to determine the origin and treatment status of blue sapphires. This approach formed the basis for selecting the Training and Testing Datasets, involving a combination of visual, spectroscopic and chemical methods and properties. The selected stones show patterns of properties consistent with those of known origin, free from any treatment—specifically, samples collected firsthand at the source. Therefore, we believe our assumption of ground truth labels is reasonably robust and suitable for the intended purpose. As an integral part of major global bodies like LMHC (Lab Manual Harmonisation Committee), which sets the standards and methods for gem labs worldwide, we adhere to globally accepted criteria. The features used to determine origin are well-known and published, rendering it unnecessary to elaborate on them here, as it goes beyond the scope and intention of this work. While the specific evaluation and weighing of criteria are typically not public knowledge, we can address the reviewer’s request by stating that the chemical elements used are published (will be made available after acceptance), and for spectroscopy, we apply the usual features known from industry publications.

Secondly, it appears that the AI model only considers blue sapphires from Kashmir, Madagascar, Burma, and Sri-Lanka. While these four locations do produce the majority of high-quality blue sapphires, other regions like Montana, Thailand, Tanzania, and Columbia also yield blue sapphires. The author should explain how the AI model performs when evaluating samples from origins beyond the main four. Misidentifying gemstones from these regions could result in significant errors and damage the laboratory’s reputation. This paper may oversimplify the challenges faced by gemological laboratories in the field.

Re: We appreciate the reviewer for highlighting this important aspect. While Gemtelligence’s approach is flexible to handle diverse geological origins, our experimental efforts have focused on metamorphic sapphires, as they represent the most common type of high-quality sapphire stone. Our training data comprises only stones from the main four origins, consequently if the network encounters a stone outside this group, its prediction will be always inaccurate due to its inherent constraints to those four origins. However, this type of error can be easily circumvented by using anomaly detection techniques that can identify whether stones presented to Gemtelligence are outliers or fall within the training set’s distribution. We have added a section in Supplementary Note 5 to detail and validate this process. It involves training and integrating a simple pre-classifier upstream of Gemtelligence to determine whether a stone belongs to the same distribution or not. The method performs effectively and, particularly in conjunction with the high threshold mode, enables the detection of

incorrectly classified stones, even when dealing with stones originating from origin outside Kashmir, Madagascar, Burma, and Sri-Lanka.

Thirdly, the author seems to exaggerate the time required for origin identification. In straightforward cases, like an obviously unheated Sri Lanka sapphire with characteristic inclusions, the process may take only a few minutes when examined under a microscope, without the need for advanced testing like FTIR, UV-VIS, or ICP-MS. Several hours of analysis may only be necessary in situations where characteristic inclusions are absent, and additional time is consumed by gemologists seeking multiple opinions, including spectroscopy analysis.

Re: Indeed, the origin of a portion of blue sapphires can be reliably determined based on microscopic characteristics. This method is reasonable for traders and other practitioners. However, professional gem labs of international recognition must not rely on a single analytical method but must collect multimodal data to corroborate an initial hypothesis. Consequently, for a professional gem lab that has to comply with best practice benchmarks when testing a high-value gemstone, the time required to acquire and evaluate different data types is not feasible within a few minutes only, even when observing a classic, textbook-like inclusion scenery.

Finally, while using deep learning for heat treatment detection can reduce reliance on human interpretation and save time, its value compared to spectral analysis algorithms, especially given existing criteria based on the presence of peaks and absorption bands[2,3], should be carefully considered.

Re: We express our gratitude to the reviewer for the comment. We would like to emphasize that the proposed deep learning algorithm considers the spectral characteristics, including the typical peaks of the sample. To further illustrate this point, we have added a new section titled “Comparison of Gemtelligence and Analytical Methods for Heat Treatment” to the main body of the manuscript. In this section, we demonstrate that well-established FTIR peaks in the gemological literature are also utilized by Gemtelligence to determine whether a gemstone has been heat-treated. Additionally, for interested readers, we have included a tool in our codebase (available with the submission) for further analysis of how the decision-making process of Gemtelligence is influenced by FTIR and UV peaks. It is essential to highlight that the algorithm’s primary value lies in combining various data sources and not in replacing spectral analysis.

Reviewer 3

In this manuscript, “Gemtelligence,” a novel automation model has been proposed to evaluate the origin and heat treatment analysis of blue sapphire. This is due to the limitations of traditional analysis, which requires experienced gemologists and multiple analytical techniques to reach the final decision. The main idea of this model is compelling and tends to be efficiently applied for origin determination and heat treatment analysis. In addition, it is a topic that is gaining

attention in the gem industry, and recently, a similar model was introduced for rubies.

Re: We thank the reviewer for acknowledging the novelty of our paper and the compelling nature of our idea. We agree that the topic is attracting the attention of the gemological industry and that several advancements have to be expected in the near future.

However, my only concern is that the same article has been published on the author's profile at <https://www.researchgate.net/>, which everyone can easily access via the link https://www.researchgate.net/publication/371490185_Gemelligence_Accelerating_Gemstone_classification_with_Deep_Learning. Is this considered self-plagiarism? Could the authors explain a difference between the already-published article and this submitted article?

Re: Thanks for raising this point. The article you are referring to is not published anywhere. It is solely the preprint version of the present paper that we uploaded to Arxiv. To the best of the authors' knowledge this is allowed by Nature Communications Engineering.

Author Responses

January 2024

General comment

We would like to thank all the reviewers and the editor for taking the time to read the manuscript. We are very glad to hear that all the reviewers recognized the potential of our approach. We are grateful for the questions we received and we believe that, by successfully addressing them and modifying the paper accordingly, the manuscript is now significantly improved. In the following, we address the concerns of each reviewer.

Reviewer 1

Deep learning shows strong potential for applications in processing large amounts of data and image recognition, and is an effective tool for performing gemstone identification in the future. This research creatively proposes a new gemstone identification system GEMTELLIGENCE that can quickly and accurately identify the origin of jewelry and determine the heat treatment. It saves the cost of testing and labor for traditional identification. There have been many attempts to apply machine learning and deep learning to gemstones in the past, the one breakthrough in this research is that the data used to train the model is not limited to tabular data, but also includes image data. But there are several issues that the authors need to take into consideration.

Re: We would like to thank the reviewer for taking the time to read our manuscript. We appreciate that you find our work creative and that you recognize the ability of Gemtelligence to process multiple modalities, beyond only tabular data. Below, we aim to address your raised questions and concerns about the paper:

1. Using multiple types of data for deep learning, how to ensure that the model can maintain a balance between different data types instead of being biased to types with large amounts of data, such as images.

Re: Thank you for raising this point. To prevent the model from becoming biased towards more abundant or informative data types, we employed random masking during training. Particularly, each data source in a training sample was randomly masked with a probability of 0.5. In a preliminary ablation study, we

found this technique to be crucial for training a model that can robustly work even when specific data sources are unavailable. We would also like to clarify that our model does not process image data. Instead, Gemtelligence processes two data types: 1) spectral data obtained from UV and FTIR spectroscopy measurements and 2) tabular data acquired from ED-XRF and ICP analyses. Supplementary Table 2, shows the specific amount of data available for training from the different types. While integrating an additional image encoder to Gemtelligence would have been straightforward and interesting from a scientific perspective, we omitted this study due to the lack of good quality image data from the stones.

2. An application case may give the reader a deeper understanding of how GEMTELLIGENCE works.

Re: Thank you for providing this valuable suggestion. We have developed a web app demo that allows readers to interact with the model’s decision-making process and gain an intuitive understanding of its capabilities. It is included with our submission material, and we would like to invite the reviewer to explore it. This demo together and the entire code base, will also be released open source after the review process.

3. More background information on the data is needed, such as where the stones were harvested from and the type of host rock, so that the reader has a good understanding of the data.

Re: We thank the reviewer for this suggestion. We have added a more comprehensive explanation of the stones’ harvesting locations, the types of host rock, and additional details of the identification process. These details can now be found in the “Materials” section and Supplementary Note 6.

4. You should give an explanation of the results of deep learning from a gemological perspective, not just a mathematical approach. You can make a feature importance analysis, or something else. For example, for sapphire, you can give a gemological interpretation of the results in terms of predictions for different origins.

Re: Thank you for your valuable advice. Inspired by your suggestion, we have decided to incorporate a new section titled “Comparison of Gemtelligence and Analytical Methods for Heat Treatment” into the manuscript. In this section, we assess whether well-established FTIR peaks in the gemological literature used for Heat Treatment Detection are also important for Gemtelligence via a counterfactual explanation approach. Our finding shows that Gemtelligence is particularly sensitive to these well established FTIR peaks, such as 3309 and 3232 cm^{-1} . We have chosen to focus on FTIR and heat treatment due to the analysis’s relative simplicity. Key FTIR bands are easily recognizable and the binary classification nature of Heat Treatment Detection makes our analysis easy to understand. However, for readers with more gemological expertise, we

have released the counterfactual explanation tool used for the analysis in our codebase, which works with both UV and FTIR data. We believe this tool will further assist interested readers in understanding Gemtelligence’s decision-making process.

Reviewer 2

In this paper, the author proposes a deep learning-assisted gemstone origin determination method for distinguishing blue sapphires from Kashmir, Madagascar, Burma, and Sri-Lanka, as well as heat treatment identification. The primary aim is to introduce automation in data analysis, ultimately reducing both time and equipment costs associated with decision-making in gemological laboratories. The author asserts that the trained AI model can confidently determine gemstone origin without relying on trace element analysis, such as laser-ablation inductively-coupled plasma mass spectrometry (ICP-MS), by solely utilizing information from infrared absorption (FTIR), UV-VIS absorption (UV-VIS), and x-ray fluorescence (XRF). Furthermore, for heat treatment identification, the AI model can achieve similar accuracy using only UV and FTIR data. The AI model significantly accelerates data analysis, reducing the time required from several hours to mere seconds and eliminating the need for expensive ICP-MS analysis in origin determination.

Re: We would like to thank the reviewer for taking the time to read our manuscript. We are happy that the reviewer found that our work can be useful to the gemological community.

In my opinion, origin determination requires more robust scientific evidence to support the author’s claims. Presently, even the most up-to-date research struggles to confidently pinpoint the origin of a gemstone based solely on a combination of spectroscopy, trace-element, and microscopy analyses. It’s important to note that the proposed AI model merely serves as a replacement for the current identification criteria employed by one gemological laboratory and does not necessarily lead to more accurate determinations. As a result, I do not recommend publishing this paper in Communications Engineering.

Re: We recognize the reviewer’s concern about the inherent complexity of origin determination and agree that it is not always possible to definitively assign an origin to all stones. However, it is important to note that for metamorphic sapphires, there exist characteristic inclusions that strongly support a certain origin, especially for Kashmir, Burma, and Sri-Lanka. Moreover, trace-elemental methods offers high degree of accuracy. Therefore, origin assignment is not feasible only for a small subset of metamorphic stones, such as those lacking characteristic inclusions or for which trace-elemental analyses yield conflicting or inconclusive results.

To avoid any potential confusion for the readers, we have included a new section titled “A Note on Ground Truth” in the manuscript. This section clarifies how the ground truths were obtained, the rationale behind this approach, and

the steps taken to further enhance our confidence in the test set’s reliability. We also acknowledge the reviewer’s observation that the stones were analyzed from a single laboratory organization. However, the OD process is based on a well-established method in which stones are compared against reference stones, which have clear provenance, making it independent of the laboratory. Moreover, the methodology we propose is not limited to data solely collected from our laboratory, and it can be applied to datasets from different labs. In addition to the new section “A Note on Ground Truth”, we have also added a section named “Limitations”. In this section, we acknowledge that our determinations are currently based on data from a single laboratory and that the usage of datasets from multiple laboratories could potentially increase the accuracy and reliability of Gemtelligence.

Overall, the proposed method demonstrates its ability to successfully differentiate between blue sapphires from Kashmir, Madagascar, Burma, and Sri-Lanka and detect heat treatment. However, concerns arise regarding the ground truth of gemstone origin in this paper, which relies on laboratory analysis rather than confirmation from the mine or the original source. While the AI model may replace existing origin determination criteria used by the Gubelin lab, it may lack sufficient scientific evidence for origin determination. Even with the latest trace element analysis, the distribution of trace elements among different origins still heavily overlaps.

Re: We thank the reviewer for raising this point. As explained in the previous response, we took stringent measures to ensure the reliability of our ground truth data. For TD, we only included stones where two independent experts reached the same conclusion regarding treatment based on visual inspection. For OD, we further required that ICP-MS also confirmed this conclusion. Since both methods rely on comparisons with reference stones and widely accepted criteria, our test set consists of stones with a minimal risk of unreliable labels. We would also like to emphasize that the inherent uncertainty of the ground truth labels is not exclusive to our study and is also present in other scientific fields such as medical diagnosis <https://www.nature.com/articles/d41586-020-00847-2>, or even within the discipline of gemology itself, e.g. color authenticity for green diamonds. Furthermore, a slight difference between the assumed and effective ground truth does not invalidate the achievement of this work, which is to demonstrate that Gemtelligence can achieve predictive performance comparable to ICP-MS and visual inspection by using only spectral methods. As mentioned in the previous response, we have added two new sections, “A Note on Ground Truth” and “Limitations”, to provide readers with more details on this crucial aspect and avoid any confusion.

The author mentions that opinions on origin remain subjective, and using biased opinions as a training model may lead to biased outcomes. The author should revise their claim to emphasize that the AI model primarily streamlines the time-consuming analysis protocol used by the Gubelin lab, rather than overemphasizing its accuracy or reliability due to the questionable training ground truth.

Re: We are grateful for the reviewer’s valuable suggestion. This paper presents compelling evidence that Gemtelligence outperforms human experts (see Figure 2) when analyzing gemstones using UV, FTIR, and XRF spectral data. The ground truth for these analyses was established using ICP and Microscopy. Beside adding the the sections mentioned above, we also have further refined the abstract and main body of the paper to eliminate any ambiguity regarding this point and to enhance clarity for readers.

Additionally, crucial details about how the ground truth, initial opinions of origin, and heat treatment were determined are missing, such as the utilization of spectroscopy and trace element analysis features. Consequently, there is a lack of scientific evidence supporting the proper generation of the ground truth.

Re: The Gübelin Gem Lab solely applies well-established and published methods and criteria to determine the origin and treatment status of blue sapphires. This approach formed the basis for selecting the Training and Testing Datasets, involving a combination of visual, spectroscopic and chemical methods and properties. The selected stones show patterns of properties consistent with those of known origin, free from any treatment—specifically, samples collected firsthand at the source. Therefore, we believe our assumption of ground truth labels is reasonably robust and suitable for the intended purpose. As an integral part of major global bodies like LMHC (Lab Manual Harmonisation Committee), which sets the standards and methods for gem labs worldwide, we adhere to globally accepted criteria. The features used to determine origin are well-known and published, rendering it unnecessary to elaborate on them here, as it goes beyond the scope and intention of this work. While the specific evaluation and weighing of criteria are typically not public knowledge, we can address the reviewer’s request by stating that the chemical elements used are published (will be made available after acceptance), and for spectroscopy, we apply the usual features known from industry publications.

Secondly, it appears that the AI model only considers blue sapphires from Kashmir, Madagascar, Burma, and Sri-Lanka. While these four locations do produce the majority of high-quality blue sapphires, other regions like Montana, Thailand, Tanzania, and Columbia also yield blue sapphires. The author should explain how the AI model performs when evaluating samples from origins beyond the main four. Misidentifying gemstones from these regions could result in significant errors and damage the laboratory’s reputation. This paper may oversimplify the challenges faced by gemological laboratories in the field.

Re: We appreciate the reviewer for highlighting this important aspect. While Gemtelligence’s approach is flexible to handle diverse geological origins, our experimental efforts have focused on metamorphic sapphires, as they represent the most common type of high-quality sapphire stone. Our training data comprises only stones from the main four origins, consequently if the network encounters a stone outside this group, its prediction will be always inaccurate due to its inherent constraints to those four origins. However, this type of error can be easily circumvented by using anomaly detection techniques that can identify

whether stones presented to Gemtelligence are outliers or fall within the training set’s distribution. We have added a section in Supplementary Note 5 to detail and validate this process. It involves training and integrating a simple pre-classifier upstream of Gemtelligence to determine whether a stone belongs to the same distribution or not. The method performs effectively and, particularly in conjunction with the high threshold mode, enables the detection of incorrectly classified stones, even when dealing with stones originating from origin outside Kashmir, Madagascar, Burma, and Sri-Lanka.

Thirdly, the author seems to exaggerate the time required for origin identification. In straightforward cases, like an obviously unheated Sri Lanka sapphire with characteristic inclusions, the process may take only a few minutes when examined under a microscope, without the need for advanced testing like FTIR, UV-VIS, or ICP-MS. Several hours of analysis may only be necessary in situations where characteristic inclusions are absent, and additional time is consumed by gemologists seeking multiple opinions, including spectroscopy analysis.

Re: Indeed, the origin of a portion of blue sapphires can be reliably determined based on microscopic characteristics. This method is reasonable for traders and other practitioners. However, professional gem labs of international recognition must not rely on a single analytical method but must collect multimodal data to corroborate an initial hypothesis. Consequently, for a professional gem lab that has to comply with best practice benchmarks when testing a high-value gemstone, the time required to acquire and evaluate different data types is not feasible within a few minutes only, even when observing a classic, textbook-like inclusion scenery. For clarity, we’ve revised the manuscript to specify that the discussed timeframe for stone analysis applies to professional gem laboratories.

Finally, while using deep learning for heat treatment detection can reduce reliance on human interpretation and save time, its value compared to spectral analysis algorithms, especially given existing criteria based on the presence of peaks and absorption bands[2,3], should be carefully considered.

Re: We express our gratitude to the reviewer for the comment. We would like to emphasize that the proposed deep learning algorithm considers the spectral characteristics, including the typical peaks of the sample. To further illustrate this point, we have added a new section titled “Comparison of Gemtelligence and Analytical Methods for Heat Treatment” to the main body of the manuscript. In this section, we demonstrate that well-established FTIR peaks in the gemological literature are also utilized by Gemtelligence to determine whether a gemstone has been heat-treated. Additionally, for interested readers, we have included a tool in our codebase (available with the submission) for further analysis of how the decision-making process of Gemtelligence is influenced by FTIR and UV peaks. It is essential to highlight that the algorithm’s primary value lies in combining various data sources and not in replacing spectral analysis.

Reviewer 3

In this manuscript, "Gemtelligence," a novel automation model has been proposed to evaluate the origin and heat treatment analysis of blue sapphire. This is due to the limitations of traditional analysis, which requires experienced gemologists and multiple analytical techniques to reach the final decision. The main idea of this model is compelling and tends to be efficiently applied for origin determination and heat treatment analysis. In addition, it is a topic that is gaining attention in the gem industry, and recently, a similar model was introduced for rubies.

Re: We thank the reviewer for acknowledging the novelty of our paper and the compelling nature of our idea. We agree that the topic is attracting the attention of the gemological industry and that several advancements have to be expected in the near future.

However, my only concern is that the same article has been published on the author's profile at <https://www.researchgate.net/>, which everyone can easily access via the link https://www.researchgate.net/publication/371490185_Gemtelligence_Accelerating_Gemstone_classification_with_Deep_Learning. Is this considered self-plagiarism? Could the authors explain a difference between the already-published article and this submitted article?

Re: Thanks for raising this point. The article you are referring to is not published anywhere. It is solely the preprint version of the present paper that we uploaded to Arxiv. To the best of the authors' knowledge this is allowed by Nature Communications Engineering.

Reviewers' comments:

Reviewer #2 (Remarks to the Author):

Review of COMMS-23-0385, Gemtelligence: Accelerating Gemstone classification with Deep Learning.

In the revision of this manuscript, the author explains the concerns from the reviewers: the ground truth of the origin data and the limitation of the method.

Although from the scientific point of view, Gemtelligence does not provide insightful information for origin and heat treatment determination and is still based on the industrial standard, the AI model effectively reduces the time consuming protocol of origin and treatment detection process applied in the Gübelin Gem Lab. The reviewer agrees that this paper presents a well trained AI model to support the decision making process in the Gübelin Gem Lab.

One minor concern is the authors did not explain how to use the confidence of the estimation. In my opinion, it is important to quantify the confidence value since the current industrial method still relies on the gemologist's objective judgment. It will be great if the authors can introduce the confidence score in the final report and let the clients understand the limitation of the origin and heat-treatment detection. In addition, how to interpolate the confidence value from the AI model into the real world prediction accuracy? For example, is 80% of prediction confidence sufficient or acceptable to be presented in the lab report? Or it is still required to be reviewed and finalized by a gemologist.

Reviewer #3 (Remarks to the Author):

The additional information provided in the revised manuscript is concise and also gives a clearer picture of how Gemtelligence works. The application related to the origin determination of blue sapphire would be useful in gemstone analysis. However, for heat treatment analysis, the authors should be more careful when using the FTIR peaks due to the many factors that affect the appearance of the peaks, such as the direction of measurement and sample orientation. In addition, the original procedure for heat treatment analysis, such as internal features and inclusion, should also be considered.

I also reviewed the paper on behalf of Reviewer 1 as asked by the Editor. The authors give a clearer overview of the system as recommended by reviewer 1. The author must ensure that the types (spectral, tabular, or images) of the data used are more clearly specified and should be in the section explaining the acquisition of the data used in the training (may be in a Data Sources section?).

Reviewer 1 has recommended that the authors explain the results from a gemological perspective regarding the deep learning model, which seems to focus on a mathematical point of view. However, the authors chose to open a new section titled "Comparison of GEMTELLIGENCE and Gemological Expert Methods for Heat Treatment" which I do not think relates to the reviewer originally questioned. I suggest that the authors just give some case studies of the blue sapphire sample with the unknown origin and make a comparative prediction from the gemologist and the

GEMTELLIGENCE to show how much more accurate their model is than the experts.

Author Responses

April 21, 2024

Revision 1

General comment

We would like to thank all the reviewers and the editor for taking the time to read the manuscript. We are very glad to hear that all the reviewers recognized the potential of our approach. We are grateful for the questions we received and we believe that, by successfully addressing them and modifying the paper accordingly, the manuscript is now significantly improved. In the following, we address the concerns of each reviewer.

Reviewer 1

Deep learning shows strong potential for applications in processing large amounts of data and image recognition, and is an effective tool for performing gemstone identification in the future. This research creatively proposes a new gemstone identification system GEMTELLIGENCE that can quickly and accurately identify the origin of jewelry and determine the heat treatment. It saves the cost of testing and labor for traditional identification. There have been many attempts to apply machine learning and deep learning to gemstones in the past, the one breakthrough in this research is that the data used to train the model is not limited to tabular data, but also includes image data. But there are several issues that the authors need to take into consideration.

Re: We would like to thank the reviewer for taking the time to read our manuscript. We appreciate that you find our work creative and that you recognize the ability of Gemtelligence to process multiple modalities, beyond only tabular data. Below, we aim to address your raised questions and concerns about the paper:

1. Using multiple types of data for deep learning, how to ensure that the model can maintain a balance between different data types instead of being biased to types with large amounts of data, such as images.

Re: Thank you for raising this point. To prevent the model from becoming biased towards more abundant or informative data types, we employed random masking during training. Particularly, each data source in a training sample was

randomly masked with a probability of 0.5. In a preliminary ablation study, we found this technique to be crucial for training a model that can robustly work even when specific data sources are unavailable. We would also like to clarify that our model does not process image data. Instead, Gemtelligence processes two data types: 1) spectral data obtained from UV and FTIR spectroscopy measurements and 2) tabular data acquired from ED-XRF and ICP analyses. Supplementary Table 2, shows the specific amount of data available for training from the different types. While integrating an additional image encoder to Gemtelligence would have been straightforward and interesting from a scientific perspective, we omitted this study due to the lack of good quality image data from the stones.

2. An application case may give the reader a deeper understanding of how GEMTELLIGENCE works.

Re: Thank you for providing this valuable suggestion. We have developed a web app demo that allows readers to interact with the model’s decision-making process and gain an intuitive understanding of its capabilities. It is included with our submission material, and we would like to invite the reviewer to explore it. This demo together and the entire code base, will also be released open source after the review process.

3. More background information on the data is needed, such as where the stones were harvested from and the type of host rock, so that the reader has a good understanding of the data.

Re: We thank the reviewer for this suggestion. We have added a more comprehensive explanation of the stones’ harvesting locations, the types of host rock, and additional details of the identification process. These details can now be found in the “Materials” section and Supplementary Note 6.

4. You should give an explanation of the results of deep learning from a gemological perspective, not just a mathematical approach. You can make a feature importance analysis, or something else. For example, for sapphire, you can give a gemological interpretation of the results in terms of predictions for different origins.

Re: Thank you for your valuable advice. Inspired by your suggestion, we have decided to incorporate a new section titled “Comparison of Gemtelligence and Analytical Methods for Heat Treatment” into the manuscript. In this section, we assess whether well-established FTIR peaks in the gemological literature used for Heat Treatment Detection are also important for Gemtelligence via a counterfactual explanation approach. Our finding shows that Gemtelligence is particularly sensitive to these well established FTIR peaks, such as 3309 and 3232 cm^{-1} . We have chosen to focus on FTIR and heat treatment due to the analysis’s relative simplicity. Key FTIR bands are easily recognizable and the binary classification nature of Heat Treatment Detection makes our analysis

easy to understand. However, for readers with more gemological expertise, we have released the counterfactual explanation tool used for the analysis in our codebase, which works with both UV and FTIR data. We believe this tool will further assist interested readers in understanding Gemtelligence’s decision-making process.

Reviewer 2

In this paper, the author proposes a deep learning-assisted gemstone origin determination method for distinguishing blue sapphires from Kashmir, Madagascar, Burma, and Sri-Lanka, as well as heat treatment identification. The primary aim is to introduce automation in data analysis, ultimately reducing both time and equipment costs associated with decision-making in gemological laboratories. The author asserts that the trained AI model can confidently determine gemstone origin without relying on trace element analysis, such as laser-ablation inductively-coupled plasma mass spectrometry (ICP-MS), by solely utilizing information from infrared absorption (FTIR), UV-VIS absorption (UV-VIS), and x-ray fluorescence (XRF). Furthermore, for heat treatment identification, the AI model can achieve similar accuracy using only UV and FTIR data. The AI model significantly accelerates data analysis, reducing the time required from several hours to mere seconds and eliminating the need for expensive ICP-MS analysis in origin determination.

Re: We would like to thank the reviewer for taking the time to read our manuscript. We are happy that the reviewer found that our work can be useful to the gemological community.

In my opinion, origin determination requires more robust scientific evidence to support the author’s claims. Presently, even the most up-to-date research struggles to confidently pinpoint the origin of a gemstone based solely on a combination of spectroscopy, trace-element, and microscopy analyses. It’s important to note that the proposed AI model merely serves as a replacement for the current identification criteria employed by one gemological laboratory and does not necessarily lead to more accurate determinations. As a result, I do not recommend publishing this paper in Communications Engineering.

Re: We recognize the reviewer’s concern about the inherent complexity of origin determination and agree that it is not always possible to definitively assign an origin to all stones. However, it is important to note that for metamorphic sapphires, there exist characteristic inclusions that strongly support a certain origin, especially for Kashmir, Burma, and Sri-Lanka. Moreover, trace-elemental methods offers high degree of accuracy. Therefore, origin assignment is not feasible only for a small subset of metamorphic stones, such as those lacking characteristic inclusions or for which trace-elemental analyses yield conflicting or inconclusive results.

To avoid any potential confusion for the readers, we have included a new section titled “A Note on Ground Truth” in the manuscript. This section clarifies how the ground truths were obtained, the rationale behind this approach, and

the steps taken to further increase our confidence in the test set’s reliability. We also acknowledge the reviewer’s observation that the stones were analyzed from a single laboratory organization. However, the OD process is based on a well-established method in which stones are compared against reference stones, which have clear provenance, making it independent of the laboratory. Moreover, the methodology we propose is not limited to data solely collected from our laboratory, and it can be applied to datasets from different labs. In addition to the new section “A Note on Ground Truth”, we have also added a section named “Limitations”. In this section, we acknowledge that our determinations are currently based on data from a single laboratory and that the usage of datasets from multiple laboratories could potentially increase the accuracy and reliability of Gemtelligence.

Overall, the proposed method demonstrates its ability to successfully differentiate between blue sapphires from Kashmir, Madagascar, Burma, and Sri-Lanka and detect heat treatment. However, concerns arise regarding the ground truth of gemstone origin in this paper, which relies on laboratory analysis rather than confirmation from the mine or the original source. While the AI model may replace existing origin determination criteria used by the Gubelin lab, it may lack sufficient scientific evidence for origin determination. Even with the latest trace element analysis, the distribution of trace elements among different origins still heavily overlaps.

Re: We thank the reviewer for raising this point. As explained in the previous response, we took stringent measures to ensure the reliability of our ground truth data. For TD, we only included stones where two independent experts reached the same conclusion regarding treatment based on visual inspection. For OD, we further required that ICP-MS also confirmed this conclusion. Since both methods rely on comparisons with reference stones and widely accepted criteria, our test set consists of stones with a minimal risk of unreliable labels. We would also like to emphasize that the inherent uncertainty of the ground truth labels is not exclusive to our study and is also present in other scientific fields such as medical diagnosis <https://www.nature.com/articles/d41586-020-00847-2>, or even within the discipline of gemology itself, e.g. color authenticity for green diamonds. Furthermore, a slight difference between the assumed and effective ground truth does not invalidate the achievement of this work, which is to demonstrate that Gemtelligence can achieve predictive performance comparable to ICP-MS and visual inspection by using only spectral methods. As mentioned in the previous response, we have added two new sections, “A Note on Ground Truth” and “Limitations”, to provide readers with more details on this crucial aspect and avoid any confusion.

The author mentions that opinions on origin remain subjective, and using biased opinions as a training model may lead to biased outcomes. The author should revise their claim to emphasize that the AI model primarily streamlines the time-consuming analysis protocol used by the Gubelin lab, rather than overemphasizing its accuracy or reliability due to the questionable training ground truth.

Re: We are grateful for the reviewer’s valuable suggestion. This paper presents compelling evidence that Gemtelligence outperforms human experts (see Figure 2) when analyzing gemstones using UV, FTIR, and XRF spectral data. The ground truth for these analyses was established using ICP and Microscopy. Beside adding the the sections mentioned above, we also have further refined the abstract and main body of the paper to eliminate any ambiguity regarding this point and to enhance clarity for readers.

Additionally, crucial details about how the ground truth, initial opinions of origin, and heat treatment were determined are missing, such as the utilization of spectroscopy and trace element analysis features. Consequently, there is a lack of scientific evidence supporting the proper generation of the ground truth.

Re: The Gübelin Gem Lab solely applies well-established and published methods and criteria to determine the origin and treatment status of blue sapphires. This approach formed the basis for selecting the Training and Testing Datasets, involving a combination of visual, spectroscopic and chemical methods and properties. The selected stones show patterns of properties consistent with those of known origin, free from any treatment—specifically, samples collected firsthand at the source. Therefore, we believe our assumption of ground truth labels is reasonably robust and suitable for the intended purpose. As an integral part of major global bodies like LMHC (Lab Manual Harmonisation Committee), which sets the standards and methods for gem labs worldwide, we adhere to globally accepted criteria. The features used to determine origin are well-known and published, rendering it unnecessary to elaborate on them here, as it goes beyond the scope and intention of this work. While the specific evaluation and weighing of criteria are typically not public knowledge, we can address the reviewer’s request by stating that the chemical elements used are published (will be made available after acceptance), and for spectroscopy, we apply the usual features known from industry publications.

Secondly, it appears that the AI model only considers blue sapphires from Kashmir, Madagascar, Burma, and Sri-Lanka. While these four locations do produce the majority of high-quality blue sapphires, other regions like Montana, Thailand, Tanzania, and Columbia also yield blue sapphires. The author should explain how the AI model performs when evaluating samples from origins beyond the main four. Misidentifying gemstones from these regions could result in significant errors and damage the laboratory’s reputation. This paper may oversimplify the challenges faced by gemological laboratories in the field.

Re: We appreciate the reviewer for highlighting this important aspect. While Gemtelligence’s approach is flexible to handle diverse geological origins, our experimental efforts have focused on metamorphic sapphires, as they represent the most common type of high-quality sapphire stone. Our training data comprises only stones from the main four origins, consequently if the network encounters a stone outside this group, its prediction will be always inaccurate due to its inherent constraints to those four origins. However, this type of error can be easily circumvented by using anomaly detection techniques that can identify

whether stones presented to Gemtelligence are outliers or fall within the training set’s distribution. We have added a section in Supplementary Note 5 to detail and validate this process. It involves training and integrating a simple pre-classifier upstream of Gemtelligence to determine whether a stone belongs to the same distribution or not. The method performs effectively and, particularly in conjunction with the high threshold mode, enables the detection of incorrectly classified stones, even when dealing with stones originating from origin outside Kashmir, Madagascar, Burma, and Sri-Lanka.

Thirdly, the author seems to exaggerate the time required for origin identification. In straightforward cases, like an obviously unheated Sri Lanka sapphire with characteristic inclusions, the process may take only a few minutes when examined under a microscope, without the need for advanced testing like FTIR, UV-VIS, or ICP-MS. Several hours of analysis may only be necessary in situations where characteristic inclusions are absent, and additional time is consumed by gemologists seeking multiple opinions, including spectroscopy analysis.

Re: Indeed, the origin of a portion of blue sapphires can be reliably determined based on microscopic characteristics. This method is reasonable for traders and other practitioners. However, professional gem labs of international recognition must not rely on a single analytical method but must collect multimodal data to corroborate an initial hypothesis. Consequently, for a professional gem lab that has to comply with best practice benchmarks when testing a high-value gemstone, the time required to acquire and evaluate different data types is not feasible within a few minutes only, even when observing a classic, textbook-like inclusion scenery. For clarity, we’ve revised the manuscript to specify that the discussed timeframe for stone analysis applies to professional gem laboratories.

Finally, while using deep learning for heat treatment detection can reduce reliance on human interpretation and save time, its value compared to spectral analysis algorithms, especially given existing criteria based on the presence of peaks and absorption bands[2,3], should be carefully considered.

Re: We express our gratitude to the reviewer for the comment. We would like to emphasize that the proposed deep learning algorithm considers the spectral characteristics, including the typical peaks of the sample. To further illustrate this point, we have added a new section titled “Comparison of Gemtelligence and Analytical Methods for Heat Treatment” to the main body of the manuscript. In this section, we demonstrate that well-established FTIR peaks in the gemological literature are also utilized by Gemtelligence to determine whether a gemstone has been heat-treated. Additionally, for interested readers, we have included a tool in our codebase (available with the submission) for further analysis of how the decision-making process of Gemtelligence is influenced by FTIR and UV peaks. It is essential to highlight that the algorithm’s primary value lies in combining various data sources and not in replacing spectral analysis.

Reviewer 3

In this manuscript, “Gemtelligence,” a novel automation model has been proposed to evaluate the origin and heat treatment analysis of blue sapphire. This is due to the limitations of traditional analysis, which requires experienced gemologists and multiple analytical techniques to reach the final decision. The main idea of this model is compelling and tends to be efficiently applied for origin determination and heat treatment analysis. In addition, it is a topic that is gaining attention in the gem industry, and recently, a similar model was introduced for rubies.

Re: We thank the reviewer for acknowledging the novelty of our paper and the compelling nature of our idea. We agree that the topic is attracting the attention of the gemological industry and that several advancements have to be expected in the near future.

However, my only concern is that the same article has been published on the author’s profile at <https://www.researchgate.net/>, which everyone can easily access via the link https://www.researchgate.net/publication/371490185_Gemtelligence_Accelerating_Gemstone_classification_with_Deep_Learning. Is this considered self-plagiarism? Could the authors explain a difference between the already-published article and this submitted article?

Re: Thanks for raising this point. The article you are referring to is not published anywhere. It is solely the preprint version of the present paper that we uploaded to Arxiv. To the best of the authors’ knowledge this is allowed by Nature Communications Engineering.

Revision 2

Reviewer 2

In the revision of this manuscript, the author explains the concerns from the reviewers: the ground truth of the origin data and the limitation of the method.

Re: We thank the reviewer for taking the time to read the revised manuscript. We are pleased to hear that the revisions address the previous reviewer’s concerns about the data origin and method limitations.

Although from the scientific point of view, Gemtelligence does not provide insightful information for origin and heat treatment determination and is still based on the industrial standard, the AI model effectively reduces the time consuming protocol of origin and treatment detection process applied in the Gübelin Gem Lab. The reviewer agrees that this paper presents a well trained AI model to support the decision making process in the Gübelin Gem Lab.

Re: We appreciate the reviewer’s acknowledgment about how our method can streamline and enhance the process of origin determination and heat treatment detection by using a deep learning model. However, we would like to emphasize that our method also offers valuable insights into the power of spectroscopy data (UV, XRF, FTIR) for the analysis of sapphires. Our findings demonstrate that

the combined analysis of UV and XRF data can achieve accuracy comparable to well-established methods like microscopy and ICP-MS for origin determination of a large number of stones. Additionally, we found that heat treatment detection can be largely achieved using only UV and FTIR data. While we agree with the reviewer that our results are based on stones analysed in a single gemological laboratory, and we acknowledge that further evaluation with stones from different gemological labs would strengthen them, we believe that our findings still offer value to the gemological community.

One minor concern is the authors did not explain how to use the confidence of the estimation. In my opinion, it is important to quantify the confidence value since the current industrial method still relies on the gemologist's objective judgment. It will be great if the authors can introduce the confidence score in the final report and let the clients understand the limitation of the origin and heat-treatment detection. In addition, how to interpolate the confidence value from the AI model into the real world prediction accuracy? For example, is 80% of prediction confidence sufficient or acceptable to be presented in the lab report? Or it is still required to be reviewed and finalized by a gemologist.

Re: We thank the reviewer for raising this point. As detailed in the section *Confidence-thresholding procedure* of our manuscript, we have implemented a calibration procedure that links the model's confidence to real world prediction accuracy. This procedure determines a network confidence threshold based on a pre-defined real-world accuracy level. Predictions exceeding this threshold are expected to have a real-world accuracy higher than the defined level, assuming novel stones share the same distribution as the validation stones used during the calibration. In our manuscript, we applied this confidence procedure to determine Mode 1 and Mode 2 thresholds, corresponding to desired accuracies of 98% and 99%, respectively. Real-world applications may necessitate a different threshold based on the desired balance between the number of stones automated and the expected real-world accuracy.

Reviewer 3

The additional information provided in the revised manuscript is concise and also gives a clearer picture of how Gemtelligence works. The application related to the origin determination of blue sapphire would be useful in gemstone analysis. However, for heat treatment analysis, the authors should be more careful when using the FTIR peaks due to the many factors that affect the appearance of the peaks, such as the direction of measurement and sample orientation. In addition, the original procedure for heat treatment analysis, such as internal features and inclusion, should also be considered.

Re: We appreciate the reviewer's positive comments on the clarity and conciseness of our revised manuscript. We agree with the reviewer's second point that single FTIR spectra peaks alone are not the sole indicators for heat treatment, and our findings corroborate this. As shown in the main experimental section, Gemtelligence's performance in detecting heat treatment improves when FTIR

is combined with UV spectroscopy. Furthermore, our experiments consider the influence of different sample orientations. When multiple FTIR spectra from various orientations are available, the final prediction is determined by averaging the individual predictions. We believe the reviewer’s comment likely refers to the newly introduced section “Comparison of Gemtelligence and gemological experts for heat treatment”. This section was added in response to Reviewer 1’s request for a gemological interpretation of the results. The goal of this section was to demonstrate Gemtelligence’s sensitivity to some well-established peaks and to show that the results align with those assessed by domain experts, thereby making the approach interpretable. We intentionally simplified the analysis by focusing on only two peaks and holding all other variables constant to ensure clarity for non-gemological professionals. However, the section is purely a qualitative analysis and a simplification, performed for demonstration purposes. We have revised the text to further clarify this point and avoid confusion. Additionally, we also moved the section in the supplementary.

I also reviewed the paper on behalf of Reviewer 1 as asked by the Editor. The authors give a clearer overview of the system as recommended by reviewer 1. The author must ensure that the types (spectral, tabular, or images) of the data used are more clearly specified and should be in the section explaining the acquisition of the data used in the training (may be in a Data Sources section?).

Re: We thank the reviewer for taking the time to also addressing the concerns of Reviewer 1 and for acknowledging that our revised manuscript gives a clear overview of the system. We would like to emphasize that our manuscript contains already a detailed description of the data used in training called **Data Sources** in the supplement, where we detail the acquisition and pre-processing steps done for each analytical data source. The reviewer can find this section at page 12.

Reviewer 1 has recommended that the authors explain the results from a gemological perspective regarding the deep learning model, which seems to focus on a mathematical point of view. However, the authors chose to open a new section titled “Comparison of GEMTELLIGENCE and Gemological Expert Methods for Heat Treatment” which I do not think relates to the reviewer originally questioned. I suggest that the authors just give some case studies of the blue sapphire sample with the unknown origin and make a comparative prediction from the gemologist and the GEMTELLIGENCE to show how much more accurate their model is than the experts.

Re: We thank the reviewer for this suggestion. The new section of our article was meant to provide a qualitative overview of how the network is sensitive to some well-established peaks and to show that the results align with those assessed by domain experts, thereby making the approach interpretable. Following the reviewer’s suggestions, we have opted to 1) Move the section “Comparison of Gemtelligence and gemological experts for heat treatment” to the supplement. 2) Add a new section in the supplement named “Examples of the practical application in the gem testing laboratory” (page 3 in the Supplementary Notes)

where we report two real use-cases of how Gemtelligence contributes to the daily workflow of the laboratory.

REVIEWERS' COMMENTS:

Reviewer #2 (Remarks to the Author):

Review of COMMSENG-23-0385C, Gemtelligence: Accelerating Gemstone classification with Deep Learning.

In the revision of this manuscript, the author explains the concerns from the reviewers: the ground truth of the origin data and the limitation of the method. The reviewer agrees that this paper presents a well trained AI model to support the decision making process and can benefit the gemology society.

Author Responses

May 17, 2024

Revision 1

General comment

We would like to thank all the reviewers and the editor for taking the time to read the manuscript. We are very glad to hear that all the reviewers recognized the potential of our approach. We are grateful for the questions we received and we believe that, by successfully addressing them and modifying the paper accordingly, the manuscript is now significantly improved. In the following, we address the concerns of each reviewer.

Reviewer 1

Deep learning shows strong potential for applications in processing large amounts of data and image recognition, and is an effective tool for performing gemstone identification in the future. This research creatively proposes a new gemstone identification system GEMTELLIGENCE that can quickly and accurately identify the origin of jewelry and determine the heat treatment. It saves the cost of testing and labor for traditional identification. There have been many attempts to apply machine learning and deep learning to gemstones in the past, the one breakthrough in this research is that the data used to train the model is not limited to tabular data, but also includes image data. But there are several issues that the authors need to take into consideration.

Re: We would like to thank the reviewer for taking the time to read our manuscript. We appreciate that you find our work creative and that you recognize the ability of Gemtelligence to process multiple modalities, beyond only tabular data. Below, we aim to address your raised questions and concerns about the paper:

1. Using multiple types of data for deep learning, how to ensure that the model can maintain a balance between different data types instead of being biased to types with large amounts of data, such as images.

Re: Thank you for raising this point. To prevent the model from becoming biased towards more abundant or informative data types, we employed random masking during training. Particularly, each data source in a training sample was

randomly masked with a probability of 0.5. In a preliminary ablation study, we found this technique to be crucial for training a model that can robustly work even when specific data sources are unavailable. We would also like to clarify that our model does not process image data. Instead, Gemtelligence processes two data types: 1) spectral data obtained from UV and FTIR spectroscopy measurements and 2) tabular data acquired from ED-XRF and ICP analyses. Supplementary Table 2, shows the specific amount of data available for training from the different types. While integrating an additional image encoder to Gemtelligence would have been straightforward and interesting from a scientific perspective, we omitted this study due to the lack of good quality image data from the stones.

2. An application case may give the reader a deeper understanding of how GEMTELLIGENCE works.

Re: Thank you for providing this valuable suggestion. We have developed a web app demo that allows readers to interact with the model’s decision-making process and gain an intuitive understanding of its capabilities. It is included with our submission material, and we would like to invite the reviewer to explore it. This demo together and the entire code base, will also be released open source after the review process.

3. More background information on the data is needed, such as where the stones were harvested from and the type of host rock, so that the reader has a good understanding of the data.

Re: We thank the reviewer for this suggestion. We have added a more comprehensive explanation of the stones’ harvesting locations, the types of host rock, and additional details of the identification process. These details can now be found in the “Materials” section and Supplementary Note 6.

4. You should give an explanation of the results of deep learning from a gemological perspective, not just a mathematical approach. You can make a feature importance analysis, or something else. For example, for sapphire, you can give a gemological interpretation of the results in terms of predictions for different origins.

Re: Thank you for your valuable advice. Inspired by your suggestion, we have decided to incorporate a new section titled “Comparison of Gemtelligence and Analytical Methods for Heat Treatment” into the manuscript. In this section, we assess whether well-established FTIR peaks in the gemological literature used for Heat Treatment Detection are also important for Gemtelligence via a counterfactual explanation approach. Our finding shows that Gemtelligence is particularly sensitive to these well established FTIR peaks, such as 3309 and 3232 cm^{-1} . We have chosen to focus on FTIR and heat treatment due to the analysis’s relative simplicity. Key FTIR bands are easily recognizable and the binary classification nature of Heat Treatment Detection makes our analysis

easy to understand. However, for readers with more gemological expertise, we have released the counterfactual explanation tool used for the analysis in our codebase, which works with both UV and FTIR data. We believe this tool will further assist interested readers in understanding Gemtelligence’s decision-making process.

Reviewer 2

In this paper, the author proposes a deep learning-assisted gemstone origin determination method for distinguishing blue sapphires from Kashmir, Madagascar, Burma, and Sri-Lanka, as well as heat treatment identification. The primary aim is to introduce automation in data analysis, ultimately reducing both time and equipment costs associated with decision-making in gemological laboratories. The author asserts that the trained AI model can confidently determine gemstone origin without relying on trace element analysis, such as laser-ablation inductively-coupled plasma mass spectrometry (ICP-MS), by solely utilizing information from infrared absorption (FTIR), UV-VIS absorption (UV-VIS), and x-ray fluorescence (XRF). Furthermore, for heat treatment identification, the AI model can achieve similar accuracy using only UV and FTIR data. The AI model significantly accelerates data analysis, reducing the time required from several hours to mere seconds and eliminating the need for expensive ICP-MS analysis in origin determination.

Re: We would like to thank the reviewer for taking the time to read our manuscript. We are happy that the reviewer found that our work can be useful to the gemological community.

In my opinion, origin determination requires more robust scientific evidence to support the author’s claims. Presently, even the most up-to-date research struggles to confidently pinpoint the origin of a gemstone based solely on a combination of spectroscopy, trace-element, and microscopy analyses. It’s important to note that the proposed AI model merely serves as a replacement for the current identification criteria employed by one gemological laboratory and does not necessarily lead to more accurate determinations. As a result, I do not recommend publishing this paper in Communications Engineering.

Re: We recognize the reviewer’s concern about the inherent complexity of origin determination and agree that it is not always possible to definitively assign an origin to all stones. However, it is important to note that for metamorphic sapphires, there exist characteristic inclusions that strongly support a certain origin, especially for Kashmir, Burma, and Sri-Lanka. Moreover, trace-elemental methods offers high degree of accuracy. Therefore, origin assignment is not feasible only for a small subset of metamorphic stones, such as those lacking characteristic inclusions or for which trace-elemental analyses yield conflicting or inconclusive results.

To avoid any potential confusion for the readers, we have included a new section titled “A Note on Ground Truth” in the manuscript. This section clarifies how the ground truths were obtained, the rationale behind this approach, and

the steps taken to further increase our confidence in the test set’s reliability. We also acknowledge the reviewer’s observation that the stones were analyzed from a single laboratory organization. However, the OD process is based on a well-established method in which stones are compared against reference stones, which have clear provenance, making it independent of the laboratory. Moreover, the methodology we propose is not limited to data solely collected from our laboratory, and it can be applied to datasets from different labs. In addition to the new section “A Note on Ground Truth”, we have also added a section named “Limitations”. In this section, we acknowledge that our determinations are currently based on data from a single laboratory and that the usage of datasets from multiple laboratories could potentially increase the accuracy and reliability of Gemtelligence.

Overall, the proposed method demonstrates its ability to successfully differentiate between blue sapphires from Kashmir, Madagascar, Burma, and Sri-Lanka and detect heat treatment. However, concerns arise regarding the ground truth of gemstone origin in this paper, which relies on laboratory analysis rather than confirmation from the mine or the original source. While the AI model may replace existing origin determination criteria used by the Gubelin lab, it may lack sufficient scientific evidence for origin determination. Even with the latest trace element analysis, the distribution of trace elements among different origins still heavily overlaps.

Re: We thank the reviewer for raising this point. As explained in the previous response, we took stringent measures to ensure the reliability of our ground truth data. For TD, we only included stones where two independent experts reached the same conclusion regarding treatment based on visual inspection. For OD, we further required that ICP-MS also confirmed this conclusion. Since both methods rely on comparisons with reference stones and widely accepted criteria, our test set consists of stones with a minimal risk of unreliable labels. We would also like to emphasize that the inherent uncertainty of the ground truth labels is not exclusive to our study and is also present in other scientific fields such as medical diagnosis <https://www.nature.com/articles/d41586-020-00847-2>, or even within the discipline of gemology itself, e.g. color authenticity for green diamonds. Furthermore, a slight difference between the assumed and effective ground truth does not invalidate the achievement of this work, which is to demonstrate that Gemtelligence can achieve predictive performance comparable to ICP-MS and visual inspection by using only spectral methods. As mentioned in the previous response, we have added two new sections, “A Note on Ground Truth” and “Limitations”, to provide readers with more details on this crucial aspect and avoid any confusion.

The author mentions that opinions on origin remain subjective, and using biased opinions as a training model may lead to biased outcomes. The author should revise their claim to emphasize that the AI model primarily streamlines the time-consuming analysis protocol used by the Gubelin lab, rather than overemphasizing its accuracy or reliability due to the questionable training ground truth.

Re: We are grateful for the reviewer’s valuable suggestion. This paper presents compelling evidence that Gemtelligence outperforms human experts (see Figure 2) when analyzing gemstones using UV, FTIR, and XRF spectral data. The ground truth for these analyses was established using ICP and Microscopy. Beside adding the the sections mentioned above, we also have further refined the abstract and main body of the paper to eliminate any ambiguity regarding this point and to enhance clarity for readers.

Additionally, crucial details about how the ground truth, initial opinions of origin, and heat treatment were determined are missing, such as the utilization of spectroscopy and trace element analysis features. Consequently, there is a lack of scientific evidence supporting the proper generation of the ground truth.

Re: The Gübelin Gem Lab solely applies well-established and published methods and criteria to determine the origin and treatment status of blue sapphires. This approach formed the basis for selecting the Training and Testing Datasets, involving a combination of visual, spectroscopic and chemical methods and properties. The selected stones show patterns of properties consistent with those of known origin, free from any treatment—specifically, samples collected firsthand at the source. Therefore, we believe our assumption of ground truth labels is reasonably robust and suitable for the intended purpose. As an integral part of major global bodies like LMHC (Lab Manual Harmonisation Committee), which sets the standards and methods for gem labs worldwide, we adhere to globally accepted criteria. The features used to determine origin are well-known and published, rendering it unnecessary to elaborate on them here, as it goes beyond the scope and intention of this work. While the specific evaluation and weighing of criteria are typically not public knowledge, we can address the reviewer’s request by stating that the chemical elements used are published (will be made available after acceptance), and for spectroscopy, we apply the usual features known from industry publications.

Secondly, it appears that the AI model only considers blue sapphires from Kashmir, Madagascar, Burma, and Sri-Lanka. While these four locations do produce the majority of high-quality blue sapphires, other regions like Montana, Thailand, Tanzania, and Columbia also yield blue sapphires. The author should explain how the AI model performs when evaluating samples from origins beyond the main four. Misidentifying gemstones from these regions could result in significant errors and damage the laboratory’s reputation. This paper may oversimplify the challenges faced by gemological laboratories in the field.

Re: We appreciate the reviewer for highlighting this important aspect. While Gemtelligence’s approach is flexible to handle diverse geological origins, our experimental efforts have focused on metamorphic sapphires, as they represent the most common type of high-quality sapphire stone. Our training data comprises only stones from the main four origins, consequently if the network encounters a stone outside this group, its prediction will be always inaccurate due to its inherent constraints to those four origins. However, this type of error can be easily circumvented by using anomaly detection techniques that can identify

whether stones presented to Gemtelligence are outliers or fall within the training set’s distribution. We have added a section in Supplementary Note 5 to detail and validate this process. It involves training and integrating a simple pre-classifier upstream of Gemtelligence to determine whether a stone belongs to the same distribution or not. The method performs effectively and, particularly in conjunction with the high threshold mode, enables the detection of incorrectly classified stones, even when dealing with stones originating from origin outside Kashmir, Madagascar, Burma, and Sri-Lanka.

Thirdly, the author seems to exaggerate the time required for origin identification. In straightforward cases, like an obviously unheated Sri Lanka sapphire with characteristic inclusions, the process may take only a few minutes when examined under a microscope, without the need for advanced testing like FTIR, UV-VIS, or ICP-MS. Several hours of analysis may only be necessary in situations where characteristic inclusions are absent, and additional time is consumed by gemologists seeking multiple opinions, including spectroscopy analysis.

Re: Indeed, the origin of a portion of blue sapphires can be reliably determined based on microscopic characteristics. This method is reasonable for traders and other practitioners. However, professional gem labs of international recognition must not rely on a single analytical method but must collect multimodal data to corroborate an initial hypothesis. Consequently, for a professional gem lab that has to comply with best practice benchmarks when testing a high-value gemstone, the time required to acquire and evaluate different data types is not feasible within a few minutes only, even when observing a classic, textbook-like inclusion scenery. For clarity, we’ve revised the manuscript to specify that the discussed timeframe for stone analysis applies to professional gem laboratories.

Finally, while using deep learning for heat treatment detection can reduce reliance on human interpretation and save time, its value compared to spectral analysis algorithms, especially given existing criteria based on the presence of peaks and absorption bands[2,3], should be carefully considered.

Re: We express our gratitude to the reviewer for the comment. We would like to emphasize that the proposed deep learning algorithm considers the spectral characteristics, including the typical peaks of the sample. To further illustrate this point, we have added a new section titled “Comparison of Gemtelligence and Analytical Methods for Heat Treatment” to the main body of the manuscript. In this section, we demonstrate that well-established FTIR peaks in the gemological literature are also utilized by Gemtelligence to determine whether a gemstone has been heat-treated. Additionally, for interested readers, we have included a tool in our codebase (available with the submission) for further analysis of how the decision-making process of Gemtelligence is influenced by FTIR and UV peaks. It is essential to highlight that the algorithm’s primary value lies in combining various data sources and not in replacing spectral analysis.

Reviewer 3

In this manuscript, “Gemtelligence,” a novel automation model has been proposed to evaluate the origin and heat treatment analysis of blue sapphire. This is due to the limitations of traditional analysis, which requires experienced gemologists and multiple analytical techniques to reach the final decision. The main idea of this model is compelling and tends to be efficiently applied for origin determination and heat treatment analysis. In addition, it is a topic that is gaining attention in the gem industry, and recently, a similar model was introduced for rubies.

Re: We thank the reviewer for acknowledging the novelty of our paper and the compelling nature of our idea. We agree that the topic is attracting the attention of the gemological industry and that several advancements have to be expected in the near future.

However, my only concern is that the same article has been published on the author’s profile at <https://www.researchgate.net/>, which everyone can easily access via the link https://www.researchgate.net/publication/371490185_Gemtelligence_Accelerating_Gemstone_classification_with_Deep_Learning. Is this considered self-plagiarism? Could the authors explain a difference between the already-published article and this submitted article?

Re: Thanks for raising this point. The article you are referring to is not published anywhere. It is solely the preprint version of the present paper that we uploaded to Arxiv. To the best of the authors’ knowledge this is allowed by Nature Communications Engineering.

Revision 2

Reviewer 2

In the revision of this manuscript, the author explains the concerns from the reviewers: the ground truth of the origin data and the limitation of the method.

Re: We thank the reviewer for taking the time to read the revised manuscript. We are pleased to hear that the revisions address the previous reviewer’s concerns about the data origin and method limitations.

Although from the scientific point of view, Gemtelligence does not provide insightful information for origin and heat treatment determination and is still based on the industrial standard, the AI model effectively reduces the time consuming protocol of origin and treatment detection process applied in the Gübelin Gem Lab. The reviewer agrees that this paper presents a well trained AI model to support the decision making process in the Gübelin Gem Lab.

Re: We appreciate the reviewer’s acknowledgment about how our method can streamline and enhance the process of origin determination and heat treatment detection by using a deep learning model. However, we would like to emphasize that our method also offers valuable insights into the power of spectroscopy data (UV, XRF, FTIR) for the analysis of sapphires. Our findings demonstrate that

the combined analysis of UV and XRF data can achieve accuracy comparable to well-established methods like microscopy and ICP-MS for origin determination of a large number of stones. Additionally, we found that heat treatment detection can be largely achieved using only UV and FTIR data. While we agree with the reviewer that our results are based on stones analysed in a single gemological laboratory, and we acknowledge that further evaluation with stones from different gemological labs would strengthen them, we believe that our findings still offer value to the gemological community.

One minor concern is the authors did not explain how to use the confidence of the estimation. In my opinion, it is important to quantify the confidence value since the current industrial method still relies on the gemologist's objective judgment. It will be great if the authors can introduce the confidence score in the final report and let the clients understand the limitation of the origin and heat-treatment detection. In addition, how to interpolate the confidence value from the AI model into the real world prediction accuracy? For example, is 80% of prediction confidence sufficient or acceptable to be presented in the lab report? Or it is still required to be reviewed and finalized by a gemologist.

Re: We thank the reviewer for raising this point. As detailed in the section *Confidence-thresholding procedure* of our manuscript, we have implemented a calibration procedure that links the model's confidence to real world prediction accuracy. This procedure determines a network confidence threshold based on a pre-defined real-world accuracy level. Predictions exceeding this threshold are expected to have a real-world accuracy higher than the defined level, assuming novel stones share the same distribution as the validation stones used during the calibration. In our manuscript, we applied this confidence procedure to determine Mode 1 and Mode 2 thresholds, corresponding to desired accuracies of 98% and 99%, respectively. Real-world applications may necessitate a different threshold based on the desired balance between the number of stones automated and the expected real-world accuracy.

Reviewer 3

The additional information provided in the revised manuscript is concise and also gives a clearer picture of how Gemtelligence works. The application related to the origin determination of blue sapphire would be useful in gemstone analysis. However, for heat treatment analysis, the authors should be more careful when using the FTIR peaks due to the many factors that affect the appearance of the peaks, such as the direction of measurement and sample orientation. In addition, the original procedure for heat treatment analysis, such as internal features and inclusion, should also be considered.

Re: We appreciate the reviewer's positive comments on the clarity and conciseness of our revised manuscript. We agree with the reviewer's second point that single FTIR spectra peaks alone are not the sole indicators for heat treatment, and our findings corroborate this. As shown in the main experimental section, Gemtelligence's performance in detecting heat treatment improves when FTIR

is combined with UV spectroscopy. Furthermore, our experiments consider the influence of different sample orientations. When multiple FTIR spectra from various orientations are available, the final prediction is determined by averaging the individual predictions. We believe the reviewer’s comment likely refers to the newly introduced section “Comparison of Gemtelligence and gemological experts for heat treatment”. This section was added in response to Reviewer 1’s request for a gemological interpretation of the results. The goal of this section was to demonstrate Gemtelligence’s sensitivity to some well-established peaks and to show that the results align with those assessed by domain experts, thereby making the approach interpretable. We intentionally simplified the analysis by focusing on only two peaks and holding all other variables constant to ensure clarity for non-gemological professionals. However, the section is purely a qualitative analysis and a simplification, performed for demonstration purposes. We have revised the text to further clarify this point and avoid confusion. Additionally, we also moved the section in the supplementary.

I also reviewed the paper on behalf of Reviewer 1 as asked by the Editor. The authors give a clearer overview of the system as recommended by reviewer 1. The author must ensure that the types (spectral, tabular, or images) of the data used are more clearly specified and should be in the section explaining the acquisition of the data used in the training (may be in a Data Sources section?).

Re: We thank the reviewer for taking the time to also addressing the concerns of Reviewer 1 and for acknowledging that our revised manuscript gives a clear overview of the system. We would like to emphasize that our manuscript contains already a detailed description of the data used in training called **Data Sources** in the supplement, where we detail the acquisition and pre-processing steps done for each analytical data source. The reviewer can find this section at page 12.

Reviewer 1 has recommended that the authors explain the results from a gemological perspective regarding the deep learning model, which seems to focus on a mathematical point of view. However, the authors chose to open a new section titled “Comparison of GEMTELLIGENCE and Gemological Expert Methods for Heat Treatment” which I do not think relates to the reviewer originally questioned. I suggest that the authors just give some case studies of the blue sapphire sample with the unknown origin and make a comparative prediction from the gemologist and the GEMTELLIGENCE to show how much more accurate their model is than the experts.

Re: We thank the reviewer for this suggestion. The new section of our article was meant to provide a qualitative overview of how the network is sensitive to some well-established peaks and to show that the results align with those assessed by domain experts, thereby making the approach interpretable. Following the reviewer’s suggestions, we have opted to 1) Move the section “Comparison of Gemtelligence and gemological experts for heat treatment” to the supplement. 2) Add a new section in the supplement named “Examples of the practical application in the gem testing laboratory” (page 3 in the Supplementary Notes)

where we report two real use-cases of how Gemtelligence contributes to the daily workflow of the laboratory.